# Interface-engineered nickel oxyhydroxide on carbon nanofibers for efficient urea oxidation and wastewater-to-energy conversion

Nasser A. M. Barakat[1,2]*, Ahmed Saadawi[1], Osama M. Erfan[3], Ibrahim Mustafa[4], Gaber Edris[4], Ayman Yousef[5]*

1 Chemical Engineering Department, Faculty of Engineering, Minia University, El-Minia, Egypt,
2 Chemical Engineering Department, Faculty of Engineering, Atatürk University, Erzurum, Turkey,
3 Department of Mechanical Engineering, College of Engineering, Qassim University, Buraydah, Saudi Arabia, 4 Chemical Engineering Department, Faculty of Engineering, King Abdulaziz University, Jeddah, Saudi Arabia, 5 Department of Chemical Engineering, College of Engineering and Computer Sciences, Jazan University, Jazan, Saudi Arabia

* nasbarakat@mu.edu.eg (NAMB); aymanhassan@jazanu.edu.sa (AY)

## Abstract

Urea is a common nitrogenous pollutant in agricultural and industrial wastewaters, and its electrochemical oxidation in alkaline media enables simultaneous detoxification and energy recovery. Herein, we report an interface-engineering strategy based on P,N-modified nickel–carbon nanofibers prepared via a scalable electrospinning–carbonization process using ammonium phosphate as a multifunctional precursor. During thermal treatment, phosphorus and nitrogen heteroatoms are incorporated into the carbon framework while regulating the surface chemistry of embedded nickel nanoparticles, promoting the formation of a NiOOH-ready interfacial layer. Structural analyses (XRD, SEM/TEM) confirm the formation of metallic Ni domains uniformly dispersed within turbostratic carbon nanofibers. XPS reveals the coexistence of $Ni^0$/$Ni^{+2}$/$Ni^{+3}$ species along with phosphate- and nitrogen-derived functionalities that enhance electronic modulation and redox reversibility. In 1.0 M KOH, the optimized composition exhibits a $Ni^{+2}$/$Ni^{+3}$ redox charge of ~12.5 mC.cm$^{-2}$. For urea electrooxidation (0.5 M urea), the catalyst delivers a maximum current density of ~115 mA.cm$^{-2}$ with a low onset potential of 0.37 V (vs Ag/AgCl). The apparent activation energy is 9.82 kJ.mol$^{-1}$, indicating a kinetically favorable NiOOH-mediated pathway. Chronoamperometry shows stable operation with ~70–80% current retention over 1000 s. When implemented as an anode in a membrane-less direct urea fuel cell, the material achieves a peak power density of ~1.1 W.m$^{-2}$ at 35 °C, demonstrating practical wastewater-to-energy feasibility. This work highlights how dual heteroatom modification of carbon nanofibers can regulate nickel interfacial chemistry, enabling efficient urea remediation coupled with renewable power generation.

**Data availability statement:** All relevant data are within the manuscript.

**Funding:** This work was supported by the Deanship of Graduate Studies and Scientific Research, Saudi Arabia (project number JU-202505362-DGSSR-ORA-2025 to AY).

**Competing interests:** The authors have declared that no competing interests exist.

## 1. Introduction

Carbon nanofibers (CNFs) have emerged as a versatile platform for (electro)chemical energy technologies because their high axial aspect ratio combines the advantages of nanoparticles (large surface area) with those of macroscopic fibers (continuous electron pathways and mechanical integrity) [1,2]. The one-dimensional architecture fosters rapid in-plane charge transport, provides abundant external surface for reactant access, and forms self-supporting porous mats that can be used directly as binder-free electrodes [3]. Additional benefits include chemical/thermal stability, facile integration into textiles or gas-diffusion layers, and straightforward post-functionalization. Owing to these attributes, CNFs are widely exploited as supports for functional electrocatalysts—for example, for alcohol/urea oxidation, water splitting, oxygen reduction, and $CO_2$ conversion—where the support dictates dispersion of the active phase, electron/ion percolation, and durability under bias [4,5].

Heteroatom modification further upgrades the catalytic competence of carbon. Phosphorus (P) doping introduces electron-rich P–C/P–O moieties that perturb the π network, polarize adjacent carbon atoms, and create Lewis-basic sites [6]. These effects enhance wettability, increase defect density, and strengthen metal–support interactions (e.g., P–O–M linkages), which are advantageous for anchoring transition-metal nanoparticles and for interfacial reactions that require $OH^-$ activation. P-modified carbons have delivered notable performance in oxygen electrocatalysis, metal–air batteries, and small-molecule oxidation [7,8].

Nitrogen (N) doping—in pyridinic, pyrrolic, or graphitic forms—typically raises the electronic conductivity of carbon, alters charge density at neighboring carbons, and can act as an intrinsic catalytic site or a strong metal-anchoring motif. N-doped CNFs therefore excel as supports for Fe/Co/Ni-based catalysts, enabling improved dispersion, electron transfer, and resistance to sintering/corrosion in alkaline media [9,10].

Dual P&N modification is particularly appealing because it combines the electronic enrichment of N with the interfacial polarity and coordination chemistry of P [11,12]. The two dopants cooperatively tune work function, local charge distribution, and hydrophilicity, while furnishing a dense population of edge/defect sites and robust P–O–M/ N–M coordination. Such synergy is expected to facilitate $OH^-$ supply and stabilize metal nanoparticles under turnover—both pivotal for alkaline electro-oxidation [13].

Electrospinning provides an attractive alternative: it is low-cost, scalable, and high-throughput, producing continuous polymer nanofibers that are readily carbonized into conductive CNFs. Crucially, molecular precursors, metal salts, and heteroatom sources can be co-dissolved in the spinning dope, enabling *one-pot* control over composition and morphology [14]. The resulting mats are mechanically robust and can be used as free-standing electrodes or processed into inks [15].

From an application viewpoint, urea-contaminated wastewaters are generated in large volumes from fertilizer production, livestock operations, municipal wastewater, and certain industrial streams [16]. If untreated, urea hydrolyzes to ammonia and nitrate/nitrite, causing eutrophication and ecological damage [17]. Conventional treatment (biological nitrification/denitrification, breakpoint chlorination, adsorption) is

effective but energy- and infrastructure-intensive, and it wastes the chemical energy stored in urea [18,19]. Electro-oxidation has therefore emerged as a promising dual-benefit process: it enables energy recovery while simultaneously removing urea under mild, ambient conditions. In contrast to conventional heteroatom-doping strategies, the use of ammonium phosphate in the electrospinning precursor enables in-situ incorporation of both phosphorus and nitrogen functionalities, leading to a homogeneous distribution of dopants and enhanced interaction with Ni active sites.

Among anodic materials, nickel-based electrocatalysts are the workhorse for urea electro-oxidation (UOR) in alkaline media because the active NiOOH surface layer forms readily and mediates the six-electron oxidation of urea to $N_2$ and carbonate [20,21]. However, insufficient activity and site utilization remain central bottlenecks, rooted in limited electrochemically active surface area (ECSA), sluggish $Ni(OH)_2 \leftrightarrow NiOOH$ conversion at realistic potentials, and inadequate mass transport in thick catalyst layers. These limitations can be alleviated through (i) morphology tuning—for instance, dispersing Ni nanoparticles on conductive CNF supports that provide continuous electron pathways and open porosity—and/or (ii) non-metal heteroatom modification (P, N, S) that improves wetting, charge distribution, and metal–support coupling [22,23]. In this context, CNFs are an optimum support, marrying mechanical integrity, conductivity, and high surface accessibility with facile heteroatom incorporation [24].

Building on theory for heteroatom-engineered carbons, P,N co-doping creates defect-rich, polarized sites with localized states near the Fermi level, which enhance interfacial charge transfer [25]. Because P is less electronegative and larger than N, P adjacent to N perturbs the local charge/spin density of the carbon lattice via its 3p lone pairs (and low-lying 3d acceptor levels), thereby activating neighboring C–N motifs [26,27]. For alkaline urea electro-oxidation (UOR), these PN ensembles are advantageous in several, mutually reinforcing ways: (i) they strengthen Lewis-basic adsorption of $OH^-$, accelerating the $Ni(OH)_2 \rightleftarrows NiOOH$ preactivation that governs UOR kinetics [28]; (ii) they provide polar sites for urea docking and early dehydrogenation on NiOOH; (iii) they form robust P–O–Ni and N–Ni coordination, improving Ni nanoparticle dispersion and stability against agglomeration/dissolution under anodic bias; and (iv) by modulating the local electronic structure, they mitigate site poisoning by urea-derived intermediates (e.g., cyanate/carbonate/$NH_x$ species) [29].

Building on these considerations, we develop a simple electrospinning-carbonization route to nickel-decorated CNFs in which ammonium phosphate additives (mono- and di-ammonium phosphate) serve as multifunctional heteroatom sources and morphology directors within a PVA/Ni precursor solution. After calcination, the process yields Ni/CNFs bearing P- and N-derived surface functionalities (predominantly phosphate-type P with trace N), together with well-dispersed Ni nanoparticles. Moreover, it is goaled to translate the catalyst to a practical device by deploying the optimized P&N-modified Ni/CNF anode in a membrane-less direct urea fuel cell (DUFC) to demonstrate real-world viability via polarization and power-density measurements. This study thus provides a scalable, low-cost strategy to heteroatom-modified CNF-supported Ni catalysts and clarifies how phosphate-assisted surface chemistry and fiber morphology cooperate to deliver high UOR performance and meaningful DUFC output.

## 2. Materials and methods

### 2.1. Materials

Poly(vinyl alcohol) (PVA, Mw 85–124 kDa, 87–89% hydrolyzed) and nickel(II) acetate tetrahydrate (Ni(CH$_3$COO)$_2$·4H$_2$O, ≥ 98%) were obtained from Sigma-Aldrich/Merck and used as received. Ammonium-phosphate salts—monoammonium phosphate (MAP, NH$_4$H$_2$PO$_4$, ≥ 99%) and diammonium hydrogen phosphate (DAP, (NH$_4$)$_2$HPO$_4$, ≥ 99%)—were purchased from Sigma-Aldrich/Merck; ammonium phosphate (tribasic) (TAP, (NH$_4$)$_3$PO$_4$, ≥ 98%; Alfa Aesar) was used only in exploratory spinning trials. Potassium hydroxide pellets (KOH, ≥ 85%, Merck) and urea (ACS reagent, ≥ 99.0%, Sigma-Aldrich) were used to prepare alkaline and urea electrolytes. Deionized water (18.2 MΩ·cm, Milli-Q) was used throughout.

For electrode inks, poly(vinylidene fluoride) (PVDF, powder, battery grade, Solef® 6020) and N,N-dimethylformamide (DMF, anhydrous, ≥ 99.8%, Sigma-Aldrich) were employed as binder and solvent, respectively. Woven carbon cloth (plain

weave, ~ 0.3–0.4 mm thickness; e.g., Fuel Cell Store, type W0S1002) served as the anode support. A commercial Pt/C-loaded carbon cloth cathode (20 wt% Pt/C, Pt loading 0.4 mg.cm$^{-2}$ on carbon cloth; Fuel Cell Store or equivalent) was used for DUFC tests. For three-electrode measurements, a glassy carbon disk working electrode (diameter 3 mm, area 0.070714 cm$^2$; CHI Instruments), an Ag/AgCl (3.0 M KCl) reference electrode, and a graphite rod counter electrode were used. Ethanol (≥99.5%, Fisher Chemical) and isopropanol (≥99.5%, Fisher Chemical) were used for electrode cleaning when needed.

## 2.2. Catalyst preparation

Spinning solutions were prepared in deionized water at room temperature. First, PVA was dissolved to form a 10 wt% stock by dispersing 1.50 g PVA in 13.5 g water and heating at 45–50 °C under stirring for 2 h until clear, followed by cooling to 22–25 °C. Nickel(II) acetate tetrahydrate (NiAc, 1.00 g) was then dissolved in 5 ml deionized water and stirred for ~20 min to ensure complete dissolution. Ammonium phosphate was introduced to the NiAc aqueous solution as the heteroatom/morphology-directing additive at loadings referenced to the NiAc mass: for monoammonium phosphate (MAP) or diammonium phosphate (DAP) the amounts were 0.01, 0.03, 0.05, and 0.07 g to obtain 1, 3, 5, and 7 wt% relative to 1.00 g NiAc, respectively; an additive-free dope (0 wt%) served as the control. The produced solutions were mixed the prepared PVA aqueous solutions. The mixture was homogenized for ≥30 min, briefly sonicated (5–10 min) to remove microbubbles, and passed through a 0.45 µm PTFE syringe filter. For reproducible spinning, we found the dope viscosity in the 400–800 mPa·s range and conductivity of ~0.6–1.5 mS.cm$^{-1}$ (0–3 wt% MAP/DAP) to yield the most stable jets, whereas higher additive contents naturally increased conductivity and required minor spinning adjustments. Triammonium phosphate was evaluated only in preliminary trials and did not yield a stable fiber jet under our conditions.

Electrospinning was carried out on a horizontal single-needle setup at 22 ± 2 °C and 35–45% relative humidity. The dope was loaded in a 5 mL syringe equipped with a 21G stainless-steel blunt needle (inner diameter ≈0.51 mm) and dispensed at 0.7 mL.h$^{-1}$ for the control and 1 wt% formulations; for 3–7 wt% MAP/DAP the flow rate was reduced to 0.30–0.40 mL h$^{-1}$ to suppress beading. A positive high voltage of 18 kV (tuned within 16–20 kV as needed) was applied to the needle with a tip-to-collector distance of 15 cm. Fibers were collected on an aluminum-foil-wrapped rotating drum (diameter ~10 cm) operated at ~300 rpm to obtain a random non-woven mat. The as-spun mats were peeled and dried under vacuum at 60 °C (≤10 mbar) for 10 h to remove residual solvent and loosely bound volatiles.

Carbonization/calcination was performed in a quartz tube furnace under flowing argon (200 sccm). After a 30 min purge at room temperature, samples were heated at 2 °C.min$^{-1}$ to 200 °C and held for 1 h to complete dehydration and polymer stabilization, then ramped at 5 °C.min$^{-1}$ to 750 °C and held for 5 h to carbonize the PVA matrix and convert the nickel precursor in situ. The furnace was allowed to cool naturally to room temperature under argon. The resulting materials are denoted AP-free (Ni/CNFs), MAP-x, and DAP-$x$, where $x$ refers to the additive wt% relative to NiAc ($x = 1, 3, 5, 7$).

## 2.3. Electrochemical measurements

Electrochemical tests were carried out in a conventional three-electrode glass cell using a glassy carbon disk working electrode (diameter 3 mm), a Pt mesh counter electrode, and an Ag/AgCl (sat. KCl) reference electrode. Unless otherwise stated, potentials are reported vs. Ag/AgCl and currents were normalized to the electrode area. The working electrode surface was rinsed with ethanol and deionized water, dried in air, and then coated with a small piece of the as-prepared nanofiber mat or a pressed flake of the calcined material; the piece was gently pressed to ensure intimate contact. All electrolytes were prepared with 18.2 MΩ·cm water. Activation/blank experiments used 1.0 M KOH; urea oxidation employed 0.5 M urea in 1.0 M KOH unless otherwise specified. For concentration studies, urea was varied between 0.33–2.0 M at fixed 1.0 M KOH, and KOH was varied between 1.0–7.0 M at fixed 0.5 M urea. Temperature-dependent measurements (30–60 °C) were performed in a jacketed cell connected to a circulating water bath after a 10 min thermal equilibration; the electrolyte temperature was monitored with a thermocouple immersed near the working electrode.

 

Cyclic voltammetry (CV) was recorded at 50 mV s$^{-1}$ over a potential window wide enough to capture the Ni$^{+2}$/Ni$^{+3}$ transition and the ensuing catalytic wave. Prior to any kinetic analysis, electrodes were conditioned in 1.0 M KOH by repeated cycling until the redox response stabilized; unless stated, the 30$^{th}$ cycle is reported for comparison. For urea oxidation, CVs were collected in fresh electrolyte immediately after activation. Onset potentials were determined from the intersection of tangents to the non-faradaic baseline and the rising catalytic branch, or equivalently by the potential at which the current density exceeded the baseline by 0.10 mA.cm$^{-2}$; both criteria gave indistinguishable trends. Chronoamperometry (CA) was performed at fixed potentials for 1000 s; when a sequence of potentials was tested, the same electrode was held stepwise from low to high potential without replacement, as indicated in the figure captions. Where relevant, linear sweep voltammetry (LSV) was acquired at 10 mV s$^{-1}$ to construct polarization curves in the kinetic region. No post-measurement iR compensation was applied unless explicitly noted; ohmic resistance was minimized by placing the reference capillary close to the working electrode. All measurements were repeated at least in duplicate to confirm reproducibility; fresh electrolytes were prepared for each concentration/temperature series.

## 2.4. Direct urea fuel cell (DUFC) assembly and testing

Membrane-less DUFCs were assembled in a parallel-plate configuration with a 3.0 × 3.0 cm carbon-cloth anode facing a commercial Pt/C air cathode (Pt loading 0.4 mg.cm$^{-2}$) at a fixed inter-electrode gap of 0.5 cm. The anode catalyst layer was prepared as an ink containing 0.06 g of the functional nanofibers (MAP, 1 wt% relative to Ni acetate in the spinning dope), 0.12 g PVDF binder and 1.0 mL DMF; the slurry was magnetically stirred (≥1 h) until homogeneous and then cast uniformly onto carbon cloth (3 × 3 cm) using a doctor-blade drawdown. Coated electrodes were dried at 80 °C (air oven) to remove solvent and then thermally treated at 350 °C for 1 h. The total mass gain was recorded gravimetrically to track the catalyst loading; only single-side coatings were used. The Pt/C cathode was used as received and exposed directly to ambient air (air-breathing mode). The electrodes were mounted in a PTFE spacer frame that defined the 0.5 cm electrolyte gap and sealed with Viton gaskets to prevent leakage.

Cell performance was measured in a two-electrode configuration (anode as working; cathode as counter/reference) using a potentiostat/galvanostat. The liquid phase was an aqueous KOH + urea mixture containing 0.5 M urea and 1.0 M KOH (for concentration studies, urea was varied in the range used in the figures while the KOH level was kept constant). The electrolyte volume was sufficient to fully submerge the electrodes and the inter-electrode gap; all tests were performed at room temperature or at controlled temperature in a jacketed glass cell connected to a circulating bath (equilibration ≥10 min before measurement). Prior to polarization measurements the open-circuit potential (OCP) was allowed to stabilize for 5–10 min. Polarization curves were obtained by linear sweep voltammetry (LSV) from OCP to 0 V at 0.01 V.s$^{-1}$. The power density was calculated as $P = I \times V / A_{cathode}$, where I is the cell current, V the cell voltage, and $A_{cathode}$ the apparent cathode area (0.00052 m$^2$) used for normalization; current densities were likewise normalized to $A_{cathode}$. Series resistance was minimized by short leads and compact geometry; no post-measurement iR compensation was applied unless specified. For temperature-dependent studies the same procedure was followed at each setpoint after thermal equilibration. All DUFC measurements were repeated at least twice with fresh electrolyte to confirm reproducibility.

## 2.5. Characterizations

Morphology was examined by field-emission scanning electron microscopy (SEM; JSM IT2000) operated at 3–5 kV using a working distance of ~5 mm. Nanofiber mats were mounted on aluminum stubs with conductive carbon tape; no metal coating was applied to avoid obscuring surface texture. Representative fiber diameters and surface features were extracted from multiple micrographs using ImageJ; at least 200 counts were used when particle/fiber size distributions were reported.

Transmission electron microscopy (TEM) was performed on a JEOL JEM-100CX II operated at 200 kV. Samples were prepared by gently dispersing a small amount of the calcined mat in ethanol with brief (<2 min) bath sonication

and drop-casting onto lacey-carbon Cu grids (300 mesh). Bright-field TEM, selected-area electron diffraction (SAED), and high-resolution TEM (HRTEM) were acquired from the same grids. Interplanar spacings were measured from HRTEM fringe images using Digital Micrograph, and particle size statistics were compiled from low-magnification images (≥200 particles). Care was taken to minimize beam damage by limiting exposure times during HRTEM.

Powder X-ray diffraction (XRD) patterns were recorded on a JEOL (Cu Kα radiation, $\lambda = 1.5406$ Å; 40 kV, 40 mA) using a Bragg–Brentano geometry. Mats were gently ground to fine powders and spread on zero-background Si holders. Data were collected over $2\theta = 10–80°$ with a step size of 0.020° and a counting time of 0.5 s per step (equivalent scan rate ~2.4° min⁻¹). Phase identification used ICDD PDF entries for fcc-Ni (04–0850), turbostratic/graphitic carbon (41–1487), NiO (78–0429), and representative nickel phosphates (e.g., 22–1119) where applicable. Apparent Ni crystallite sizes were estimated from the (111) reflection using the Scherrer equation with K = 0.9 after subtracting instrumental broadening measured on a LaB$_6$/Si standard.

X-ray photoelectron spectroscopy (XPS) was carried out on a Thermo Scientific K-Alpha (Al Kα, $h_v = 1486.6$ eV) with a monochromated source, base pressure $<1 \times 10^{-8}$ mbar, and a 400 μm analysis spot. Survey spectra were collected at 200 eV pass energy; high-resolution C 1s, N 1s, O 1s, P 2p, and Ni 2p regions were acquired at 20–40 eV pass energy. A low-energy flood gun was used for charge compensation; all binding energies were referenced to the adventitious C 1s peak at 284.8 eV. Spectra were processed in CasaXPS using a Shirley background and mixed Gaussian–Lorentzian (Voigt) line shapes. Atomic percentages were derived from peak areas corrected by Scofield sensitivity factors and analyzer transmission. No ion sputter cleaning was applied to avoid reduction of surface nickel species; all spectra represent the as-prepared surface after brief (<1 min) exposure to ambient air prior to loading.

## 3. Results and discussion

### 3.1. Electrocatalyst characterization

The crystalline phases of the calcined nanofibers were examined by X-ray diffraction (Fig 1A). The diffractogram is dominated by the characteristic reflections of face-centered cubic (fcc) metallic nickel at $2\theta \approx 44.5°$, 51.8°, and 76.4°, indexed to the (111), (200), and (220) planes, respectively (ICDD PDF 00-004-0850). The sharp, intense Ni peaks confirm efficient reduction of the nickel precursor to crystalline Ni during calcination under inert atmosphere. A broad halo centered around $2\theta \approx 25–26°$ is assigned to the (002) stacking of graphitic carbon (ICDD PDF 41–1487), produced from PVA pyrolysis; the $d_{002}$ is slightly expanded relative to ideal graphite, indicating a turbostratic/poorly ordered carbon matrix. A weak shoulder near 42–44° can include contributions from the (100)/(101) graphite bands but overlaps with Ni(111), rendering the carbon signal less pronounced in this region.

Because the spinning solution contained ammonium phosphate, interactions between Ni$^{+2}$ and phosphate species are expected before and/or during heating. In water, the precursors dissociate as:

$$Ni(CH_3COO)_2 . 4H_2O \rightarrow Ni^{+2} + 2CH_3COO^- + 4H_2O \tag{1}$$

$$(NH_4)_2HPO_4 \leftrightarrow 2NH_4^+ + HPO_4^{-2}(/H_2PO_4^-/PO_4^{-3}, pH-dependent) \tag{2}$$

Under suitable local pH, partial precipitation/complexation can occur:

$$3Ni^{+2} + 2PO_4^{-3} + xH_2O \rightarrow Ni_3(PO_4)_2 . xH_2O(s) \tag{3}$$

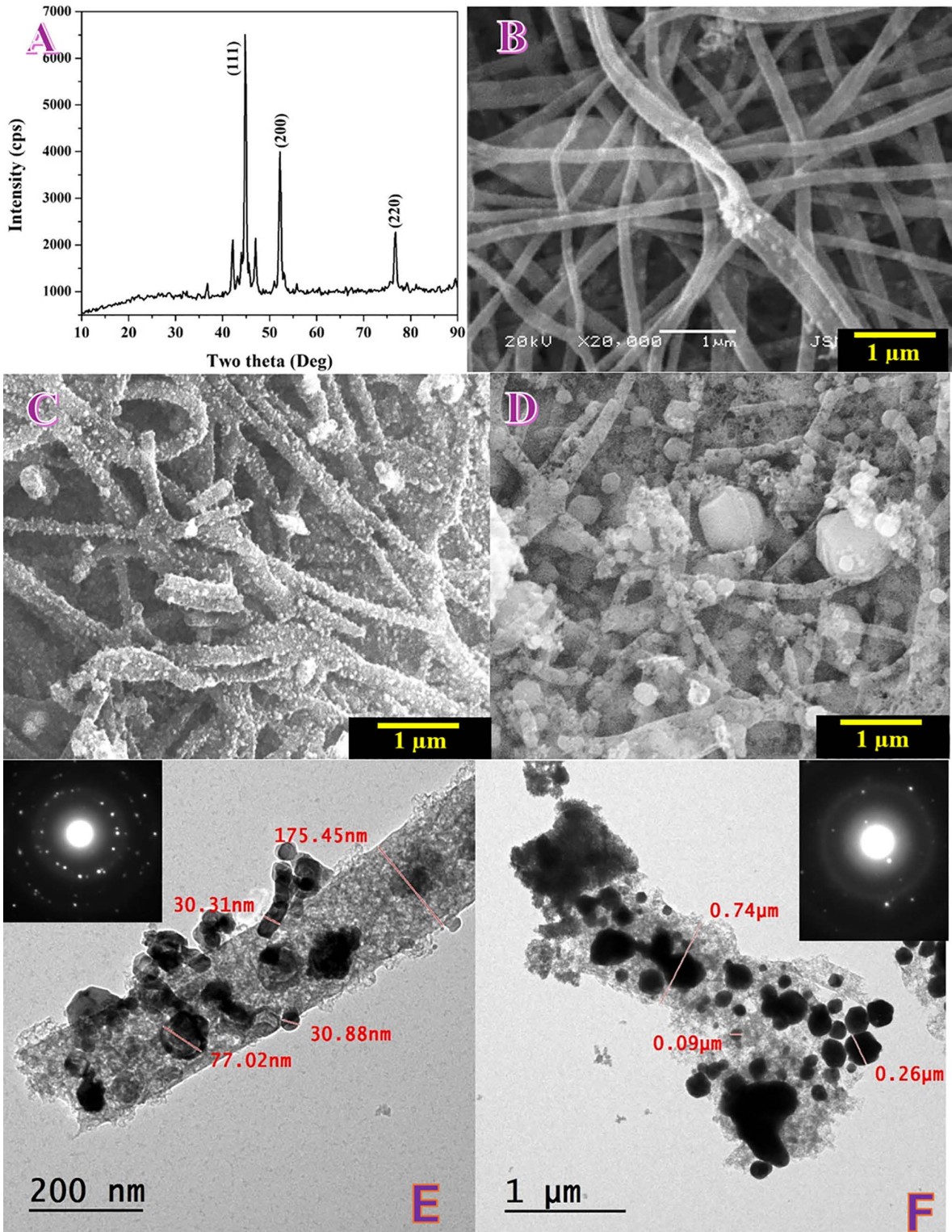

**Fig 1. Chemical composition, morphology and internal structure of the prepared nanofibers.** A) XRD pattern for the obtained powder after calcination of the PVA/NiAc/DAP electrospun mats under inert atmosphere at 750 °C. SEM images for the nanofibers obtained from calcination of different

electrospun mats: B) Ammonium phosphate-free, C) Mono ammonium phosphate-containing and D) Di ammonium phosphate-containing. Normal TEM image (E) for the nanofibers obtaining from electrospun solution containing MAP, the inset in panel displays SAED pattern. (F) normal TEM images for the nanofibers produced from electrospun solutions containing DAP, the inset is the corresponding SAED pattern.

During drying and calcination, ammonium phosphate decomposes (releasing $NH_3/H_2O$ and forming poly-/metaphosphates or $P_2O_5$), while acetate and the evolving carbonaceous species (CO, $H_2$) furnish a reducing environment. Transient nickel phosphate and/or NiO intermediates can therefore convert to metallic Ni via steps such as:

$$Ni_3(PO_4)_2 \overset{\triangle}{\rightarrow} 3NiO + P_2O_5 \tag{4}$$

The unreacted nickel acetate decomposes under inert atmosphere with this reactions sequence [30,31].

$$Ni(CH_3COO)_2 \cdot 4H_2O \rightarrow 0.86Ni(CH_3COO)_2 \cdot 0.14Ni(OH)_2 + 0.28CH_3COOH + 3.72H_2O \tag{5}$$

$$3(0.86\ Ni(CH_3COO)_2 \cdot 0.14Ni(OH)_2) + 0.28H_2O \rightarrow 3NiCO_3 + 2.44CH_3COCH_3 + 1.12H_2 \tag{6}$$

$$NiCO_3 \rightarrow NiO + CO_2 \tag{7}$$

$$NiO + CO/H_2 \rightarrow Ni + CO_2/H_2O \tag{8}$$

These pathways rationalize why the final XRD pattern is overwhelmingly metallic Ni plus carbon, while any phosphate-derived residues remain minor and likely amorphous. Consistently, only faint features in the 37–43° window—potentially from residual NiO or poorly crystalline phosphate species—are observed and are much weaker than the Ni reflections [32].

Notably, all samples prepared with MAP or DAP (0.01–0.07 g) exhibit essentially the same phase assemblage: predominant fcc-Ni embedded in a turbostratic carbon matrix, with no resolvable crystalline nickel phosphate phases after calcination. This similarity indicates that, at the employed phosphate levels and processing conditions, phosphate mainly acts through transient complexation/precipitation and decomposition chemistry that ultimately feeds the reduction of Ni salts to metallic Ni without leaving a crystalline phosphate fingerprint in the final nanofibers.

In summary, the XRD results confirm nanocrystalline fcc-Ni dispersed in a carbonaceous matrix, with carbon being poorly ordered and any phosphate- or oxide-related species present only as minor/amorphous traces. The intimate coupling of conductive carbon with metallic Ni is expected to be advantageous for electrocatalytic applications by combining abundant Ni active sites with efficient electron transport and structural support.

The phosphate-free precursor yields a uniform non-woven mat of continuous, cylindrical fibers with smooth surfaces and few defects (Fig 1B). The fibers weave into an open network with ample inter-fiber junctions and no obvious bead formation. The smoothness indicates that the viscoelasticity and charge density of the spinning dope were well balanced, allowing a stable whipping jet and solvent evaporation without local Rayleigh break-up. After calcination, this morphology is preserved, giving a coherent carbonaceous scaffold that provides electronic pathways and abundant contact points for the subsequently formed nickel domains inferred from XRD.

Introducing 1 wt% MAP (relative to Ni acetate), Fig 1C, changes the surface texture noticeably. Fibers remain continuous, but their surfaces become rough and granular, with dense, sub-100-nm nodules decorating the entire fiber length. The granular shell reflects salt–polymer–metal interactions in the jet: phosphate anions partially complex or precipitate

with Ni$^{+2}$ during drying, and these nanometric nuclei are immobilized by the PVA matrix and later converted during heat treatment to Ni/NiO–derived nanoparticles anchored on the fiber skin. The result is a "raspberry-like" coating that increases the specific surface area and is expected to expose more accessible active sites while maintaining the mechanical integrity of the fiber network.

With 1 wt% DAP, the salt effect becomes stronger. The mat still contains fibers, but their surfaces are densely encrusted with nanoparticles and small aggregates, and isolated micron-scale faceted crystallites and granular debris appear between and on the fibers. Such over-decoration is consistent with a higher effective phosphate basicity/ionic strength, which raises solution conductivity and screens polymer chain entanglement [33]. During electrospinning this promotes local jet instabilities and extensive in-flight nucleation, so a larger fraction of the inorganic phase forms as discrete particles or agglomerates rather than as finely dispersed nuclei within the polymer jet. As shown in Fig 1D, upon calcination these crystallites persist as larger Ni-bearing or phosphate-derived residues, producing a heterogeneous texture with partial loss of the sleek, cylindrical fiber contour seen in the phosphate-free sample. While this morphology maximizes roughness, the presence of coarse aggregates can reduce the fraction of electrochemically accessible surface and may impede ion transport within the mat.

The inability to electrospin the solution prepared with the fully ammoniated phosphate is in line with these trends. At that high neutralization level the dope likely exhibits excessive ionic strength and extensive salt–polymer and salt–Ni$^{+2}$ interactions, which depress the effective chain entanglement number and alter surface tension [34]. The jet then favors electrospraying/beading or breaks before solidification, preventing the formation of continuous filaments. Practically, reducing the phosphate content, increasing PVA concentration (to restore entanglement), or adjusting the solvent polarity and spinning field would be required to recover a stable fibrous jet.

In sum, phosphate acts as a powerful morphology modifier: a small amount (MAP) produces uniformly nanoparticle-decorated fibers that are advantageous for catalysis, whereas a stronger phosphate environment (DAP) drives extensive surface encrustation and secondary particle formation, approaching the limit where continuous electrospinning is no longer feasible. These observations align with the XRD-derived phase evolution, supporting a picture in which phosphate promotes in situ nucleation of Ni-containing nanoparticles that, after calcination, decorate a conductive carbon nanofiber scaffold.

A representative fiber fragment (Fig 1E, normal TEM image for MAP-based nanofibers) is shown decorated with a dense population of metal nanoparticles distributed along the surface ("necklace-like" arrangement). The polymer-derived carbon core appears as a lighter, continuous backbone on which darker particles are immobilized. Particle metrics taken from the micrograph indicate a prevalent size in the 25–40 nm range, with occasional coalesced domains ~70–80 nm and rare larger aggregates ≥150–180 nm. This hierarchical dispersion is consistent with phosphate-assisted nucleation in the electrospun jet: partial complexation/precipitation between Ni$^{+2}$ and $H_2PO_4^{-}$/$HPO_4^{-2}$ produces many nanoscale nuclei that are arrested by the PVA matrix. During calcination, decomposition of acetate/phosphate and the locally reducing gases (CO/$H_2$) convert these nuclei to metallic Ni while the carbonized PVA retains them at the fiber surface. The outcome is a high density of firmly anchored Ni nanoparticles on a conductive carbon scaffold—an architecture expected to enhance electrochemical accessibility and electron transport.

The selected-area electron diffraction pattern (inset in Fig 1E) displays concentric rings with superimposed discrete spots, evidencing a polycrystalline ensemble containing some larger, well-oriented crystallites. The ring spacings correspond to the principal planes of fcc-Ni—(111), (200), (220), and higher-order reflections—fully consistent with the XRD results. A weak, diffuse outer halo is frequently observed in such systems and can be attributed to the turbostratic carbon matrix; any crystalline nickel-phosphate or oxide residues, if present, are below the detection limit or highly disordered and therefore do not give distinct rings.

The DAP sample (Fig 1F) shows a carbon-fiber fragment densely loaded with metal particles, but the particle size distribution is much broader than in the MAP case. Besides numerous nanoparticles, there are coalesced aggregates in the sub-micron range (representative diameters labeled ≈0.09, 0.26 and 0.74 μm). This contrasts with the mono-phosphate

fibers, where most particles were 25–40 nm with only occasional ~70–80 nm domains. The coarser features here indicate enhanced in-flight/early nucleation and subsequent coalescence during drying and calcination when DAP is present—consistent with its higher basicity/ionic strength, which raises solution conductivity, screens polymer–polymer entanglements, and favors particle growth outside the confined fiber skin. The carbon backbone remains discernible as a lighter support, and the dark particles appear strongly anchored, but local smoothing of the fiber contour and the presence of isolated large crystallites suggest partial sintering of neighboring nuclei during heat treatment.

The SAED pattern (inset in Fig 1F) exhibits continuous rings with superimposed discrete spots, characteristic of a polycrystalline ensemble containing some larger, better-oriented crystallites. The ring positions correspond to fcc-Ni principal reflections—(111), (200), (220)—in agreement with XRD. Relative to MAP sample, the rings here are more spotty/segmentary, consistent with the larger crystallite size inferred from TEM (broader size distribution with sub-micron aggregates). A weak diffuse halo is also visible, attributable to the turbostratic carbon matrix; no distinct rings from crystalline phosphate phases are resolved, implying that any phosphate-derived residues are amorphous or below the SAED detection limit after calcination.

Fig 2 compares the high-resolution XPS of the mono-ammonium-phosphate (MAP) and di-ammonium-phosphate (DAP) nanofibers (all spectra charge-referenced to C 1s = 284.8 eV). Panel A (MAP, C 1s) displays three components at 284.83 eV (FWHM 1.22 eV, 64.02%, $sp^2$/$sp^3$ C–C/C–H), 285.79 eV (1.84 eV, 27.49%, C–O/C–O–C ± C–N) and 287.60 eV (3.40 eV, 8.49%, C=O/O–C=O), i.e., 36.0% oxygenated carbon. On the other hand, in DAP (Panel B) the envelope appears at 284.76 eV (1.26 eV, 69.43%), 286.05 eV (2.05 eV, 23.80%) and 288.30 eV (2.88 eV, 6.77%), reducing the oxygenated fraction to 30.6%. Notably, the DAP oxygenated peaks are upshifted relative to MAP by +0.26 eV (C–O) and +0.70 eV (C=O), consistent with a more inorganic/ligand-rich outer surface carbon matrix [35].

Panel C (MAP, O 1s) resolves three oxygen environments: 529.33 eV (1.34 eV, 21.93%) for lattice Ni–O–Ni ($NiO/NiO_x$), 531.12 eV (1.82 eV, 26.22%) for Ni–OH/NiOOH and non-bridging P=O/P–O$^-$, and 532.42 eV (2.72 eV, 51.84%) for C–O/adsorbed $H_2O$ and bridging P–O–M/P–O–C. By contrast, DAP (Panel D) is dominated by higher-BE oxygen and requires only 531.14 eV (1.45 eV, 22.58%) and 532.36 eV (2.56 eV, 77.42%); a distinct lattice $O^{-2}$ peak near 529 eV is not resolved. This points to a thicker hydrated/ligated overlayer in DAP that attenuates lattice-oxide emission.

Panel E (MAP, N 1s) shows a single oxidized-nitrogen component at 402.94 eV (FWHM 1.71 eV; N–O/N$^+$–O$^-$) with no intensity at 397–400 eV (no nitride or graphitic N) [36]. Similarly, DAP (Panel F) exhibits a single peak ≈403 eV overlapping the total fit; thus nitrogen remains only as trace oxidized surface species in both materials.

Panel G (MAP, P 2p) presents a phosphate (P$^{+5}$) doublet at 133.11/134.67 eV (FWHM 1.65/2.24 eV; 70.21/29.79%), with no signal at 129–131 eV (no phosphide). In DAP (Panel H) the phosphate envelope appears at 132.71 eV (FWHM 0.53 eV, 11.56%) and 133.29 eV (2.36 eV, 88.44%), still within the P$^{+5}$ range and again without phosphide. The slight lower-BE shift and asymmetric distribution for DAP are compatible with stronger P–O–Ni/C coordination and higher hydration of surface phosphate [37].

Panel I (MAP, Ni 2p) exhibits the expected oxidized nickel multiplet: Ni$^{+2}$ $2p_{3/2}$ = 853.85 eV (1.54 eV, 9.91%) and $2p_{1/2}$ = 871.88 eV (2.73 eV, 8.92%); Ni$^{3+}$ $2p_{3/2}$ = 856.10 eV (3.37 eV, 31.62%) and $2p_{1/2}$ = 873.97 eV (2.60 eV, 6.36%); with strong satellites at ~861 eV (22.05%), ~864.46 eV (7.96%), ~876.95 eV (5.75%), ~880.17 eV (7.43%) (total satellites ≈ 43%). On the other hand, DAP (Panel J) is markedly more Ni$^{+3}$-rich—Ni$^{+3}$ $2p_{3/2}$ = 856.05 eV (39.18%), Ni$^{+3}$ $2p_{1/2}$ = 873.23 eV (16.63%)—and even reveals a Ni$^0$ $2p_{3/2}$ at 852.88 eV (FWHM 1.31 eV, 5.15%), consistent with larger metallic cores beneath a more oxidized shell [38]. The satellite share remains high (≈39% at 861.31/865.39/876.72/881.11 eV), but the balance shifts from satellites to Ni$^{+3}$ main lines in DAP.

Taken together, the side-by-side format underscores clear, systematic differences: relative to MAP, DAP displays less oxygenated carbon (30.6% vs 36.0%), oxygen spectra dominated by hydroxyl/adsorbate/phosphate with no resolvable lattice-O$^{-2}$, phosphate that is slightly lower in BE and more strongly coordinated/hydrated, and a nickel envelope that is enriched in Ni$^{3+}$ with a small but distinct Ni$^0$ contribution. In structural terms, MAP retains a thinner oxide where lattice

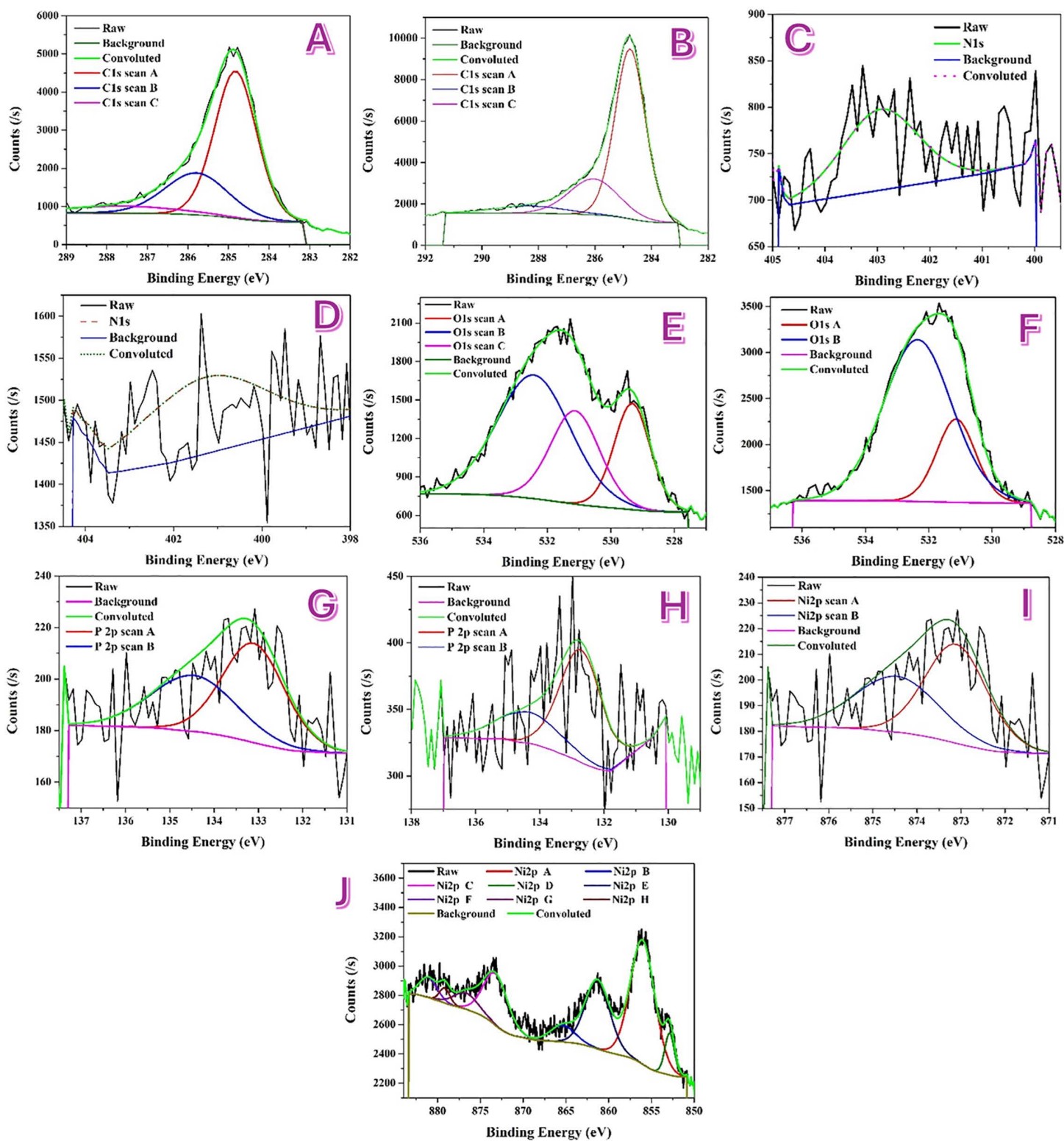

**Fig 2. XPS spectra of mono- (MAP) and di- (DAP) ammonium-phosphate incorporated Ni/CNFs.** C 1s: A) MAP and B) DAP. N 1s: C) MAP and D) DAP. O 1s: E) MAP and F) DAP. P 2p: G) MAP and H) DAP. Ni 2p: I) MAP and J) DAP.

Ni–O remains visible, whereas DAP forms Ni$^0$ cores wrapped by a thicker, NiOOH-rich and phosphate/adsorbate-laden shell—a surface chemistry expected to prime DAP for rapid Ni(OH)$_2$/NiOOH redox activation during glycerin electro-oxidation, while MAP offers a cleaner oxide/carbon balance with a detectable lattice-oxide component.

The difference in catalytic performance between MAP- and DAP-modified samples can be attributed to their distinct surface chemical environments, as revealed by XPS. Variations in phosphate and nitrogen functionalities influence the interaction with Ni active sites, thereby affecting the Ni$^{+2}$/Ni$^{+3}$ redox behavior and the stability of the NiOOH phase during electrooxidation.

### 3.2. Electrochemical activity

**3.2.1. Effect of P&N-doping.** Fig 3A compares the 30[th] cyclic voltammogram of the AP-free (Ni), MAP-containing, and DAP-containing nanofibers in 1.0 M KOH (scan range −0.2 to ~1.0 V vs Ag/AgCl). In alkaline electrolyte, nickel activates through the well-known surface redox couple [39].

$$\text{Ni(OH)}_2 \rightleftharpoons \text{NiOOH} + \text{H}^+ + \text{e}^-$$

(9)

so the magnitude and sharpness of the Ni$^{+2}$/Ni$^{+3}$ wave report the density and accessibility of electroactive Ni sites. The AP-free Ni nanofiber electrode shows the weakest redox response: broad, low-intensity Ni$^{+2}$→Ni$^{+3}$ oxidation on the forward scan and a diffuse Ni$^{+3}$→Ni$^{+2}$ reduction on the reverse scan, with a comparatively large separation between the anodic and cathodic features. This points to a small population of active Ni(OH)$_2$/NiOOH sites and slower interfacial kinetics [40].

Introducing MAP increases both the anodic and cathodic charges and narrows the redox wave, evidencing a larger number of accessible Ni sites and improved charge transfer. This behavior is fully consistent with the surface chemistry from Fig 2: MAP fibers possess a thin NiO$_x$(OH)$_y$ shell with a resolvable lattice-O component (O 1s at 529.3 eV) atop conductive turbostratic carbon (C 1s with ~36% oxygenated groups), which together favor rapid electron/ion transport and robust wetting.

The DAP sample delivers the strongest activation signal of all three: the Ni$^{+2}$/Ni$^{+3}$ wave is more intense and better defined, and the anodic current at high potentials rises most steeply. XPS explains this trend: DAP exhibits Ni$^0$ cores wrapped by a more NiOOH-rich, hydrated shell (Ni 2p dominated by Ni$^{+3}$ main lines; O 1s ruled by 531–532.4 eV hydroxyl/adsorbate/phosphate oxygen) and a slightly less oxygenated carbon surface. The hydrated, ligand-rich overlayer shortens the induction to NiOOH and increases the redoxable Ni fraction, which translates into the larger activation charge observed in CV. In short, the activation sequence follows DAP > MAP > AP-free Ni matching the hierarchy inferred from XPS of Ni oxidation state and surface oxygen speciation.

The electrochemically active surface area (ECSA) for the three formulations was estimated from the anodic Ni$^{+2}$ Ni$^{+3}$ charge using following equation [41]:

$$ECSA = \frac{1}{q.v} \int_{E_1}^{E_2} \left( j_{anode} - j_{baseline} \right) dE$$

(10)

Where $v = 0.05$ V.s$^{-1}$, $j$ is the current, $E$ is the applied potential and is the charge density range for Ni(OH)$_2$/NiOOH (~0.39 mC.cm$^{-2}$). The estimated ECSA values for the produced nanofibers was 1.285, 1.95 and 2.545 cm$^2$ for the AP-free, MAP-containing and DAP-containing nanofibers, respectively. A larger ECSA and a surface already biased toward NiOOH (DAP) should deliver a higher density of active sites and faster OH$^-$ activation during urea electro-oxidation. MAP, with a thinner oxide but good wettability (more oxygenated carbon), is intermediate, while the AP-free mat provides the fewest accessible sites. The electrochemical activation behavior in Fig 6A therefore mirrors the XPS-derived surface architecture and foreshadows the same performance order under urea oxidation conditions.

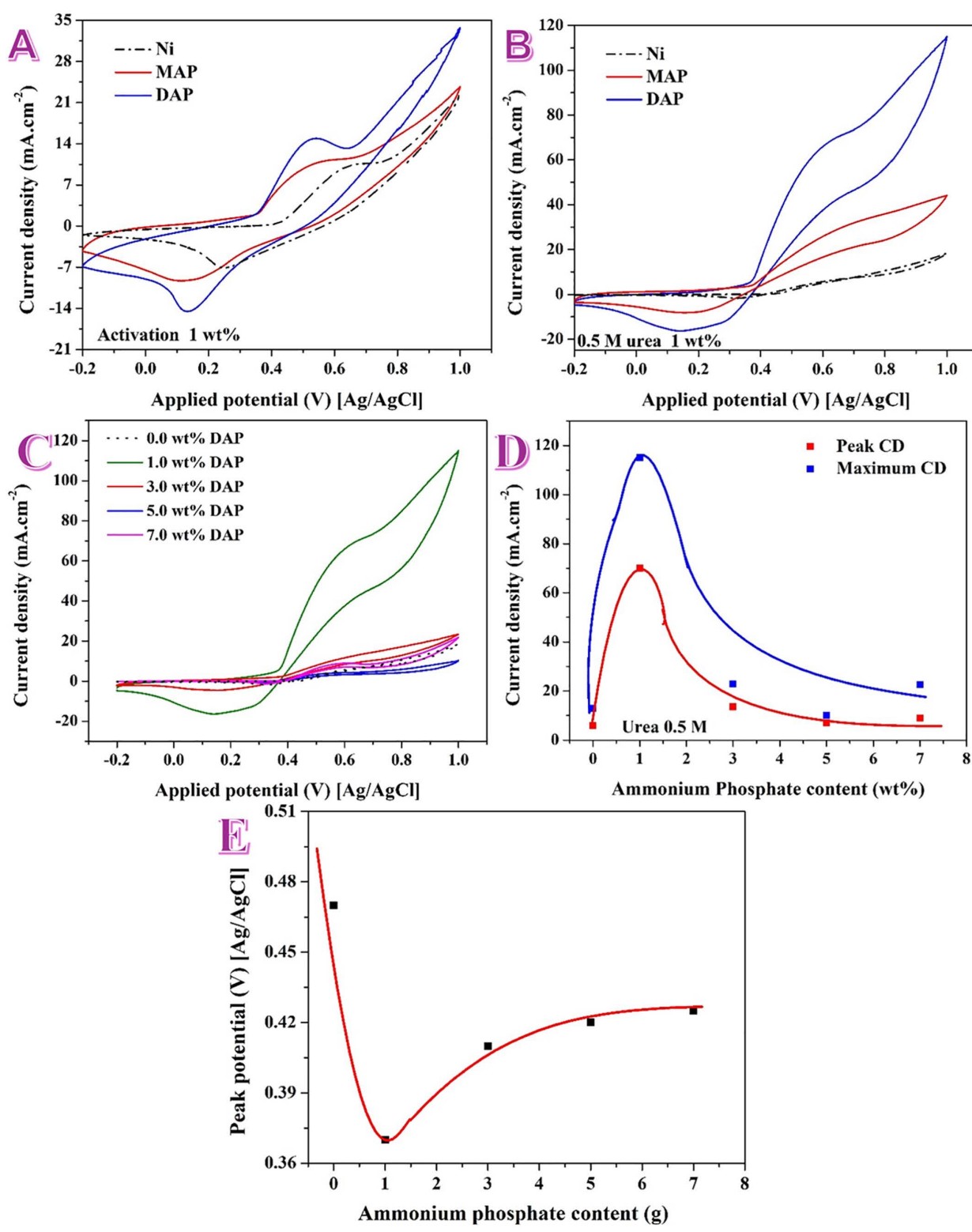

**Fig 3. Cyclic voltammetry voltammograms at 50 mV/s for the prepared nanofibers.** A) 30th cycle in presence of 1.0 M KOH, and B) in presence of 0.5 M urea solution (in 1.0 M KOH). C) cyclic voltammograms for nanofibers prepared from electrospun solutions having different contents from DAP. D) Effect of DAP on the maximum and anodic peak current density. E) Effect of DAP content on the onset potential.

Fig 3B displays the CV measurements for the three electrodes in presence of 0.5 M urea (in 1.0 M KOH). As shown, the anodic current rises strongly once NiOOH is formed. The catalytic reaction proceeds on NiOOH according to the overall 6-electron step

$$\mathbf{(NH_2)_2CO + 6OH^- \rightarrow N_2 + CO_3^{-2} + 5\,H_2O + 6e^-} \tag{11}$$

The voltammograms in Fig 3B show a clear ranking: DAP > MAP > AP-free Ni. The DAP electrode (blue) displays both the earliest onset of urea oxidation (≈0.42 V vs Ag/AgCl) and the highest current across the potential window; at 1.00 V it reaches 115 mA.cm$^{-2}$, versus 43 mA.cm$^{-2}$ for MAP and 15 mA.cm$^{-2}$ for Ni. The MAP electrode shows a negative shift of onset relative to bare Ni (≈0.36 vs ≈ 0.48 V) and a sizable current enhancement, but it remains distinctly below DAP.

The XPS results explain these electrochemical trends. Ni valence & readiness for NiOOH. DAP exhibits a Ni$^{+3}$-rich surface with a small Ni$^0$ core signal (Ni 2p: 856.05/873.23 eV dominating), while MAP shows a smaller Ni$^{+3}$ share and a thinner oxide shell. The DAP surface is therefore closer to the active NiOOH state, lowering the kinetic barrier and shifting the UOR onset negatively. Surface oxygen speciation. O 1s for DAP is dominated by 531–532.4 eV hydroxyl/adsorbate/phosphate oxygen (77%) and lacks a distinct lattice-O$^{-2}$ peak. This hydrated, ligand-rich skin supplies –OH to the interface and facilitates urea adsorption/dehydrogenation on NiOOH. MAP retains a detectable lattice-O component (529.3 eV, 22%), consistent with a thinner, less hydrated shell and slightly slower activation. Phosphate environment. Both materials show P$^{+5}$ (PO$_4$), but DAP's P 2p is slightly lower in BE and more asymmetric, consistent with stronger P–O–Ni/ P–O–C coordination. These anionic ligands help anchor Ni particles, maintain hydration, and can promote interfacial proton/oxide transfer.

Moreover, DAP presents a rougher, more heavily decorated surface (SEM/TEM) and, from the Ni redox charge in KOH (Fig 6A, 30th cycle, 50 mV/s), a larger electrochemically active surface area. Using the Ni(OH)$_2$/NiOOH charge method (details in Fig 3A discussion). The current ratio at 1.00 V (≈2.4 × DAP/MAP) exceeds the ECSA ratio (≈1.3×), implying a genuine kinetic/electronic promotion by the NiOOH-rich, phosphate-hydrated surface unique to DAP. The pronounced anodic currents for DAP persist on the reverse sweep without catastrophic decay, indicating that the active NiOOH layer is maintained under urea electrooxidation conditions and that severe passivation is not dominant in this window.

Urea electro-oxidation is governed by how readily the surface converts to—and operates as—NiOOH while supplying interfacial –OH. The DAP nanofibers offer exactly that: Ni$^0$ cores for conductivity wrapped by a hydrated, Ni$^{+3}$-rich shell decorated with phosphate—delivering the lowest onset and highest currents. MAP benefits from increased ECSA and good wettability but lags behind DAP, while bare Ni is limited by fewer active sites and slower Ni(OH)$_2$ ↔ NiOOH cycling.

Tuning the DAP content in the spinning dope strongly modulates the UOR current in 0.5 M urea + 1.0 M KOH as shown in Fig 3C. The 1 wt% DAP electrode exhibits a steep rise after the Ni(OH)$_2$ → NiOOH transition and delivers the highest currents across the potential window. Either decreasing DAP to 0 wt% (no phosphate) or increasing it to 3–7 wt% progressively damps the response and shifts the curve toward higher potentials.

As shown in Fig 3D, both the maximum current density and the anodic peak current density show a clear volcano-type dependence with an optimum at 1 wt% DAP: The maximum current density was 13 mA.cm$^{-2}$ (0%) → 115 mA.cm$^{-2}$ (1%) → 22.8 mA.cm$^{-2}$ (3%) → 10 mA.cm$^{-2}$ (5%) → 22.5 mA.cm$^{-2}$ (7%). While the anodic current density (in mA.cm$^{-2}$) was 6 (0%) → 70 (1%) → 13.6 (3%) → 7 (5%) → 9 (7%). Thus, relative to the phosphate-free baseline, 1 wt% DAP boosts the maximum current density by ~8.8× and anodic current desnity by ~11.7 ×. Raising DAP above 1 wt% sharply reduces both metrics (to ~0.2–0.3 of the optimum), with a small partial recovery at 7 wt% that remains far below the 1 wt% level.

The onset follows the same volcano trend, Fig 3E, reaching a minimum at 1 wt% DAP:

$$E_{\text{onset}}(\text{V vs Ag/AgCl}) = 0.47\ (0\%) \rightarrow 0.37\ (1\%) \rightarrow 0.41\ (3\%) \rightarrow 0.42\ (5\%) \rightarrow 0.425\ (7\%).$$

The 0.10 V negative shift from 0% to 1% DAP evidences a marked kinetic facilitation of UOR.

At low DAP (0%), XPS/SEM indicate fewer Ni nuclei and a thinner, less hydrated surface; the Ni(OH)$_2$/NiOOH redox charge ([Fig 6A]) and hence ECSA are smallest, giving low UOR currents and the most positive onset. At the optimum (1%), TEM shows dense nanoparticle decoration without excessive agglomeration; XPS shows Ni$^0$ cores wrapped by a Ni$^{+3}$-rich, hydroxylated/phosphate-ligated shell (Ni 2p dominated by Ni$^{+3}$; O 1s dominated by 531–532.4 eV OH/adsorbate/P–O). This architecture (i) maximizes electroactive surface area, (ii) supplies interfacial –OH and stabilizing P–O–Ni/P–O–C linkages, and (iii) places the surface close to the active NiOOH state—together yielding the lowest onset and highest currents.

At excess DAP (≥3%), SEM/TEM evidence over-decoration and sub-micron aggregates; the surface becomes overly hydrated/ligated and partly blocked by phosphate, decreasing ECSA and increasing mass-transport/charge-transfer resistances. The Ni shell can thicken toward an insulating hydroxide/adsorbate layer, pushing the onset positive and depressing the current despite the presence of Ni. The modest recovery at 7% likely comes from additional metallic domains that partially restore conductivity, but site accessibility remains compromised. Although higher ECSA values indicate a greater number of accessible active sites, the overall catalytic performance is also influenced by charge-transfer kinetics and surface chemistry. Therefore, the optimal activity observed for specific compositions reflects a balance between active-site density and intrinsic catalytic efficiency.

### 3.2.2. Effect of solution concentration.

Effect of urea solution concentration has been investigated using three DAP-based nanofibers; 1, 3 and 7 wt%. For the 1 wt% electrode, as shown in [Fig 4A], in blank KOH only the Ni$^{+2}$/Ni$^{+3}$ couple is observed. Adding urea (0.33→2.0 M) turns the anodic branch into a pronounced catalytic wave with a negative shift of onset (~0.05–0.1 V) and a monotonic rise of current up to ~1.0 V. Between 0.33 and 1.0 M the increase is steep; from 1.0 to 2.0 M the gain becomes modest, suggesting approach to surface-coverage/transport saturation at the most active sites. The strong concentration sensitivity is consistent with this composition's large ECSA and NiOOH-ready, hydroxylated/phosphate-ligated surface (XPS: Ni$^{+3}$-rich; O 1s dominated by 531–532.4 eV). For the 3 wt% electrode ([Fig 4B]), the same qualitative trend holds, but absolute currents are much lower and the onset shift with concentration is smaller. Catalytic currents increase with urea from 0.33 to 2.0 M, yet the curves remain widely separated from the 1 wt% case, indicating fewer accessible active sites and/or higher interfacial resistance.

Currents are the smallest of the series in case of 7 wt% electrode ([Fig 4C]); the onset is only weakly affected by urea concentration and the anodic rise is shallow even at 2.0 M. The nearly flat blank response and muted catalytic gain point to site blockage/low ECSA and partial mass-transport limitation through an insulating overlayer—again consistent with the structural picture at high DAP content.

Across all panels the UOR current scales positively with urea concentration because the active NiOOH phase (formed in situ) oxidizes urea; at low urea concentration the rate is kinetic/coverage-limited, while at high urea concentration it approaches a quasi-saturation regime. The magnitude of the concentration response follows the catalyst quality: 1 wt% DAP » 3 wt% DAP > 7 wt% DAP, mirroring the volcano dependence vs DAP loading. Increasing urea concentration enhances currents for all electrodes, but only the 1 wt% DAP catalyst converts that increase into large, early-onset catalytic waves; over-doping (3–7 wt%) dampens both the onset shift and current gain because of lower ECSA and hindered charge/mass transfer.

[Fig 4C] presents CVs of the 1 wt% DAP nanofiber electrode in 0.5 M urea while varying KOH from 1.0 to 7.0 M. A clear, nearly monotonic enhancement is observed as KOH concentration increases: the catalytic wave initiates at progressively lower potentials, the slope of the anodic branch becomes steeper, and the upper-potential current grows substantially. Between 1.0 and 3.0–4.0 M, both the onset and the current increase rapidly; above ~5.0 M, the gain persists but begins to taper, suggesting proximity to a quasi-saturation regime where charge transfer rather than substrate delivery limits the rate.

These trends are mechanistically expected for the Ni(OH)$_2$/NiOOH-mediated urea oxidation pathway. Higher OH$^-$ activity accelerates formation of NiOOH (the true active phase), shifts its appearance to lower potential, and supplies reactant

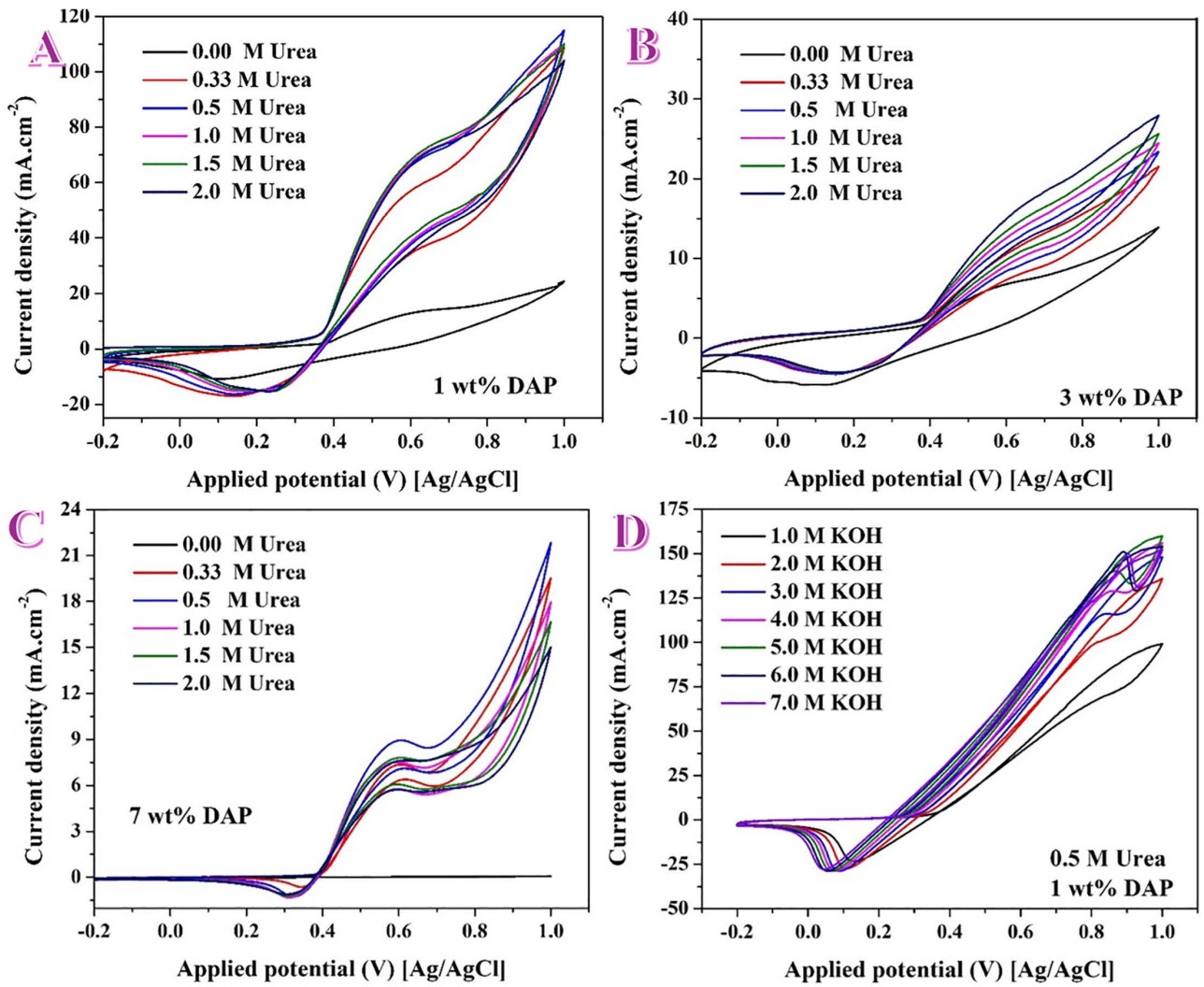

**Fig 4. Effect of urea solution concentration on the urea electrooxidation reaction over three DAP-containing electrodes:** A) **1 wt%,** B) **3 wt% and** C) **7 wt%.** D) Effect of KOH solution concentration on the urea electrooxidation process over the proposed P&N-doped Ni/CNFs (1 wt% DAP) in presence of 0.5 M urea at 50 mV/s scan rate.

to the chemical dehydrogenation steps on NiOOH. The electrolyte conductivity rises with KOH concentration, reducing ohmic drop and revealing the intrinsic kinetics more clearly. Increased ionic strength compresses the double layer, raising the near-surface concentration of urea/OH⁻.

The DAP surface chemistry established by XPS reinforces this behavior: the 1 wt% DAP fibers expose $Ni^0$ cores wrapped by a $Ni^{+3}$-rich, highly hydroxylated/phosphate-ligated shell (Ni 2p dominated by $Ni^{+3}$; O 1s dominated by 531–532.4 eV OH/adsorbate/P–O). As KOH concentration increases, this hydrated shell more readily converts to and sustains NiOOH, while phosphate linkages (P–O–Ni/P–O–C) and oxygenated groups maintain interfacial wetting and OH⁻ supply. The net result is the observed negative shift of the UOR onset and strong rise of catalytic current from 1 to 7 M KOH. Any slight leveling at the highest concentrations is consistent with the approach to surface-coverage/transport limits rather than a change of mechanism. In short, for the optimally formulated DAP catalyst, boosting KOH concentration continuously

improves urea electro-oxidation by promoting NiOOH formation, enhancing OH⁻ availability and conductivity, and leveraging the inherently NiOOH-ready surface architecture revealed by XPS.

The improved catalytic activity can be attributed to the synergistic effect of phosphorus and nitrogen functionalities introduced by ammonium phosphate. Phosphate groups facilitate OH⁻ adsorption and stabilize the NiOOH active phase, while nitrogen doping enhances electrical conductivity and strengthens metal–support interactions. These effects collectively promote the $Ni^{2+}/Ni^{3+}$ redox transformation and improve urea electrooxidation kinetics.

### 3.2.3. Effect of reaction temperature and electrode stability.

Fig 5A shows that heating the electrolyte markedly boosts the urea-oxidation, over the 1wt% DAP electrode at 50 mV/s, current and shifts the catalytic onset to more negative potentials. From 30→60 °C the anodic branch steepens continuously and the current at 1.0 V grows by roughly a factor of 2–3, while the oxidation wave appears ~0.08–0.10 V earlier. The hysteresis between forward and reverse sweeps also diminishes with temperature, indicating faster surface regeneration of the active NiOOH phase and improved removal of adsorbed intermediates.

These trends are mechanistically expected for the $Ni(OH)_2$/NiOOH-mediated pathway. Higher temperature (i) accelerates the $Ni(OH)_2 \rightarrow NiOOH$ pre-activation step, (ii) enhances the chemical dehydrogenation steps of urea on NiOOH, and (iii) increases OH⁻ mobility and solution conductivity, reducing ohmic loss [20,42]. The strong temperature sensitivity is fully consistent with the DAP surface chemistry deduced by XPS—$Ni^0$ cores wrapped by a $Ni^{+3}$-rich, highly hydroxylated/phosphate-ligated shell—which readily converts to and sustains NiOOH and supplies interfacial –OH.

Fig 5B presents an Arrhenius plot built from a pseudo-rate constant $k$ (proportional to the catalytic current measured at a fixed potential within the kinetic region). The excellent linearity of $Ln(k)$ vs $1/T$ ($R^2 = 0.95$) indicates that the same rate-limiting step governs the reaction over 30–60 °C and that mass-transport limitations are not dominating in the analyzed potential window. According to Arrhenius equation, the slope of the fit equals $-E_a/R$; thus the apparent activation energy $E_a$ follows directly from the line.

The Arrhenius analysis yields an apparent activation energy $E_a = 9.82$ kJ/mol for the urea electro-oxidation on the 1 wt% DAP nanofibers. This is a low barrier for Ni-based UOR and implies that, in the potential window used to extract $k$, the reaction is kinetically facile and only moderately temperature-dependent. Such a small $E_a$ is consistent with a surface that is already close to the active NiOOH state and richly supplied with –OH ligands, as established by XPS for this catalyst (hydroxyl/adsorbate/phosphate oxygen). In this scenario the rate-determining event is likely an early dehydrogenation/electron-transfer step on NiOOH rather than a high-barrier C–N cleavage, explaining the negative onset shift and strong performance at near-ambient temperature. We note that $E_a$ is "apparent" and depends on the chosen potential and kinetic regime; nevertheless, the good linearity indicates a single dominant mechanistic step across 30–60 °C for these optimized DAP electrodes. Overall, elevating the temperature lowers the kinetic barrier for urea electro-oxidation on the optimally formulated DAP nanofibers, producing earlier onset, larger currents, and Arrhenius behavior consistent with a single thermally activated step on a NiOOH-rich surface.

Fig 5C shows current–time transients recorded on the same 1 wt% DAP electrode at potentials from 0.40 to 1.00 V vs Ag/AgCl. At each step the current exhibits (i) a short, sharp initial drop (t < 20–40 s), and (ii) a slower approach to a quasi-steady value. The fast decay reflects double-layer charging and rapid restructuring of the surface as $Ni(OH)_2 \rightarrow NiOOH$ forms under bias. The magnitude of the initial spike scales with potential, consistent with faster growth of the active NiOOH phase at higher bias.

Quasi-steady region and stability. After ~100–150 s the currents stabilize and then decline only gradually. From visual integration of the traces (±10% error from the plot): 1.00 V: ~70→55 mA.cm⁻² at 1000 s (≈80% retention). 0.90 V: ~36→26 mA.cm⁻² (≈72% retention). 0.80 V: ~22→15 mA.cm⁻² (≈68% retention). 0.70 V: ~18→12 mA.cm⁻² (≈67% retention). 0.60 V: ~13→9 mA.cm⁻² (≈70% retention). 0.50 V: ~10→7 mA.cm⁻² (≈70% retention). 0.40 V: ~5→4 mA.cm⁻² (≈80% retention).

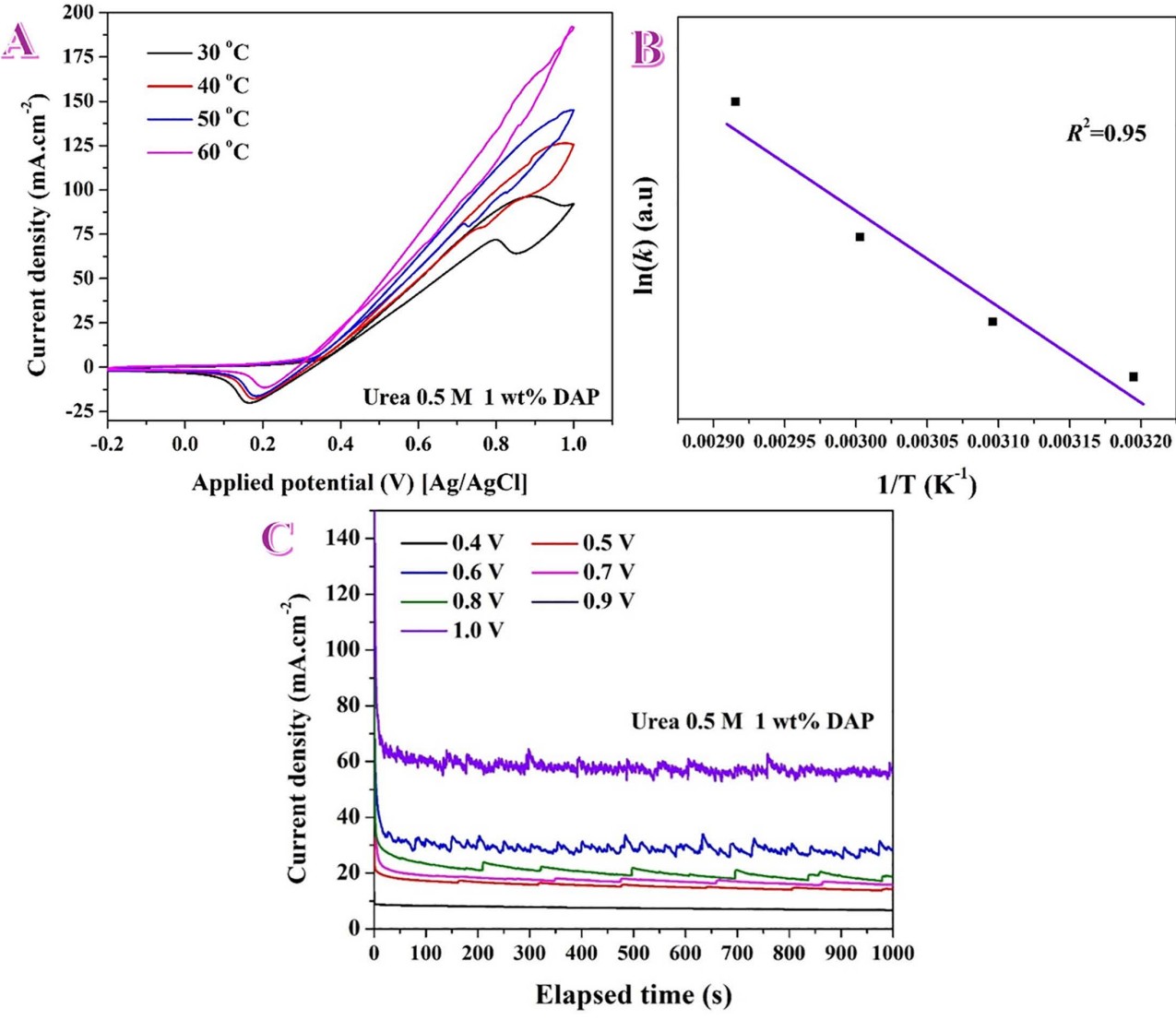

**Fig 5. Temperature-dependent electrochemical behavior and chronoamperometric stability of the 1 wt% DAP electrode toward urea electro-oxidation in alkaline medium.** A) Effect of temperature on the electrocatalytic activity of the 1 wt% DAP electrode using 0.5 M urea solution (in 1.0 KOH). B) Arrhenius plot for the obtained data. C) Chronoamperometry measurements at various potnetioals for 1 wt% DAP in presence of 0.5 M (in 1.0 M KOH).

The highest sustained current is obtained at 1.00 V, while 0.85–0.95 V offers a good compromise between current and stability (≥70% current retained at 1000 s). Small step-like drops and noise, more visible at 0.8–1.0 V, are attributed to episodic bubble coverage/detachment (UOR and concurrent OER) and the intermittent removal of adsorbed intermediates.

The gradual decay is typical for UOR on NiOOH and arises from (i) partial site blocking by reaction intermediates and carbonate, (ii) mass-transport limitations in the diffusion layer, and (iii) competition with OER at the highest potentials [43]. The ability of the DAP electrode to maintain large steady currents—especially at 0.9–1.0 V—aligns with its $Ni^{+3}$-rich, highly hydroxylated/phosphate-ligated surface established by XPS. This surface converts readily to NiOOH, supplies interfacial $OH^-$, and benefits from P–O–Ni/P–O–C linkages that stabilize Ni sites against dissolution and agglomeration, giving

sustained activity under load. Because the same electrode was used sequentially, mild conditioning of the NiOOH layer across steps is expected; nevertheless, the consistent retention across potentials indicates robust durability of the 1 wt% DAP catalyst in 0.5 M urea.

The observed stability can be attributed to the robust structural features of the catalyst, including the confinement of Ni nanoparticles within the carbon nanofiber matrix and the presence of P- and N-containing surface functionalities, which enhance metal–support interactions and suppress structural degradation during electrochemical operation.

Overall, compared with previously reported Ni-based electrocatalysts for urea oxidation, the present ammonium-phosphate-modified Ni–carbon nanofibers offer a favorable combination of a conductive fibrous carbon framework, uni-formly distributed Ni active sites, and phosphate/nitrogen surface functionalities. These features promote efficient $Ni^{+2}$/$Ni^{+3}$ redox transformation, enhance charge transfer, and improve catalyst stability during prolonged operation. The initial current decay observed during chronoamperometry can be attributed to double-layer charging and the rapid adsorption of reaction intermediates, followed by the establishment of a steady-state surface coverage and formation of the NiOOH active phase.

**3.2.4. Statistical modeling and optimization analysis.** To quantify how composition tunes the catalytic response, we used Design-Expert to build a quadratic RSM for the peak current density $j_{peak}$ extracted from CVs of the optimized DAP-nanofiber anode. The model considered two factors in coded form: the DAP content in the spinning dope (A, 0–7 wt% relative to Ni acetate) and the urea concentration in the electrolyte (B; 0–2.0 M in 1.0 M KOH). The fitted polynomial has the usual structure,

$$J_a = -378.61 - 1641.11A + 10.08B + 4.3AB - 2339.03A^2 - 14.64B^2 - 17.85A^2B - 59.94AB^2$$
$$-1337.34A^3 - 14.53B^3 - 30.8A^2B^2 - 8.74A^3B - 19.44AB^3 - 260.62A^4 - 9.67B^4$$

(12)

with linear, interaction, and curvature terms.

The 2D contour map (Fig 6A) and the 3D surface (Fig 6B) reveal a narrow high-response ridge located at low A (around 1 wt% DAP) that extends across moderate-to-high B (roughly 0.8–1.5 M urea). Movement along the DAP axis displays a clear volcano behavior: activity is maximized near 1 wt%, then declines for ≥3 wt% despite the larger heteroatom supply. This matches our electrochemical trends and XPS interpretation—excess DAP over-ligates/over-hydrates the surface, partly blocking NiOOH sites, whereas ~1 wt% DAP yields a $Ni^{+3}$-rich, hydroxylated shell that is "NiOOH-ready."

Movement along the urea axis gives a positive but saturating response: increasing urea from 0 to ~1–1.5 M raises $j_{peak}$, after which gains level off. The saturation is consistent with site coverage of urea/intermediates, increasing viscosity, and mass-transport limitations at the highest urea content. Together, the RSM points to a robust operating window around A ≈ 1 wt% and B ≈ 1.0–1.5 M, in agreement with our factor-by-factor CV study.

ANOVA confirms the model is highly significant (F = 90.4, p < 0.0001), i.e., the regression captures the dominant variability in $j_{peak}$. The lack-of-fit/ residual test is also significant (F = 18.81, p = 0.0062), which is common when the pure error from replicates is very small; practically, it flags that some higher-order effects (e.g., subtle transport or ohmic contributions at the composition extremes) are not fully represented by the quadratic. Even so, the diagnostic plots validate predictive utility:

- Fig 6C (Predicted vs. experimental/ residuals) shows points clustered tightly about the 45° line and residuals symmetri-cally distributed around zero, indicating no systematic bias across the design space.

- Fig 6D (model fit with 95% confidence/prediction bands) shows the experimental responses lying within narrow uncer-tainty envelopes, which broaden only near the edges (where data density is naturally lower). The gentle concavity along the urea axis visualizes the negative $\beta_{22}$ (saturation), while the sharper curvature along the DAP axis reflects the volcano captured by a positive $\beta_1$ and negative β11.

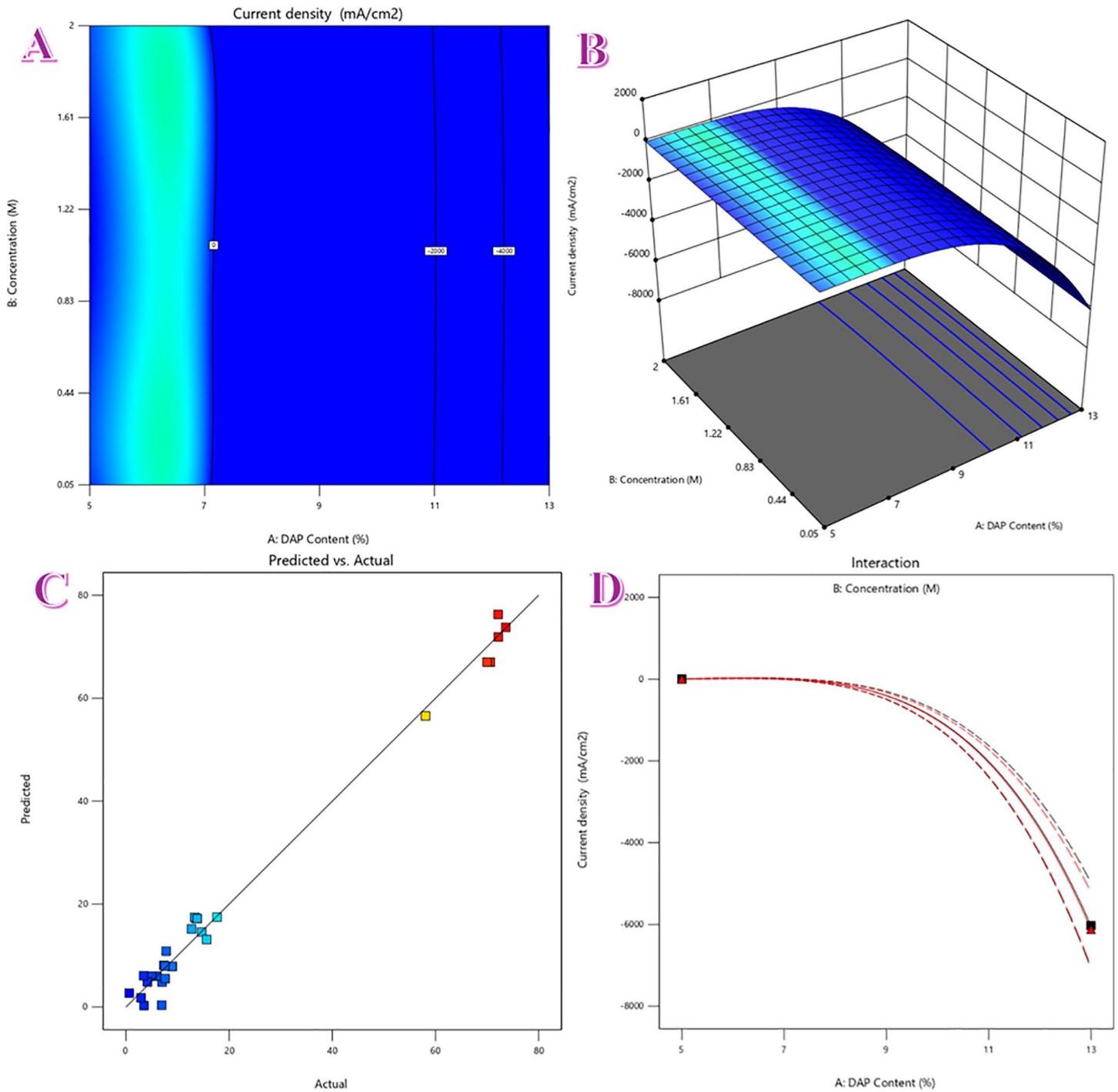

**Fig 6. Response-surface modeling of the CV peak current density $j_{peak}$ for the DAP–Ni/CNF anode.** (A) 2D contour map versus DAP content in the spinning dope and urea concentration (coded units). (B) 3D response surface showing a ridge near ~1 wt% DAP and ~1–1.5 M urea. (C) Predicted vs. experimental $j_{peak}$ with residuals indicating good parity. (D) Model fit with 95% confidence/prediction bands. ANOVA: model F = 90, p < 0.0001; lack-of-fit/residual F = 18.81, p = 0.0062.

The statistics and surfaces together formalize our mechanistic picture:

- DAP content primarily sets the density and accessibility of NiOOH sites through shell chemistry (XPS: $Ni^{+3}$ enrichment, hydroxylated/phosphate ligation). This yields a true optimum near 1 wt% DAP.

- Urea concentration feeds the surface reaction and boosts current until adsorption/transport saturation is approached around ~1–1.5 M, beyond which further increases offer little benefit.

Thus, the RSM recommends operating near 1 wt% DAP and ~1–1.5 M urea to maximize $j_{peak}$; outside this corridor, either insufficient active-site utilization (low DAP/low urea) or site blocking/transport penalties (high DAP/high urea) limit gains. The excellent model significance, supportive diagnostics, and alignment with independent electrochemical/XPS evidence provide strong confidence that these conclusions are both statistically and physically grounded.

### 3.3. Direct Urea Fuel Cell (DUFC)

Implementing the catalyst in a working DUFC is essential to translate half-cell activity into device-level metrics. We fabricated a membrane-less air-cathode cell using the MAP (1 wt%) nanofiber anode ink on carbon cloth and a commercial Pt/C air cathode; polarization curves were recorded by LSV (0.01 V.s$^{-1}$) and current/power were normalized to the cathode area (0.00052 m$^2$).

Polarization behavior (Fig 7A). The open-circuit potential (OCP) lies near 0.41–0.46 V at low urea content and decreases as the urea concentration is raised to 2.0 M (down to ~0.25–0.30 V). This drop is characteristic of membrane-less DUFCs and originates from fuel crossover to the air cathode, which creates a mixed potential and partially suppresses the ORR. Under load, three regions are evident: (i) an activation-controlled zone at low current; (ii) a quasi-linear ohmic region; and (iii) mass-transport limitation at high current [44]. Increasing urea concentration shifts the curves to higher current densities, reflecting improved fuel availability at the anode, but the concurrent OCP loss and stronger cathode polarization at high urea concentration limit the net voltage.

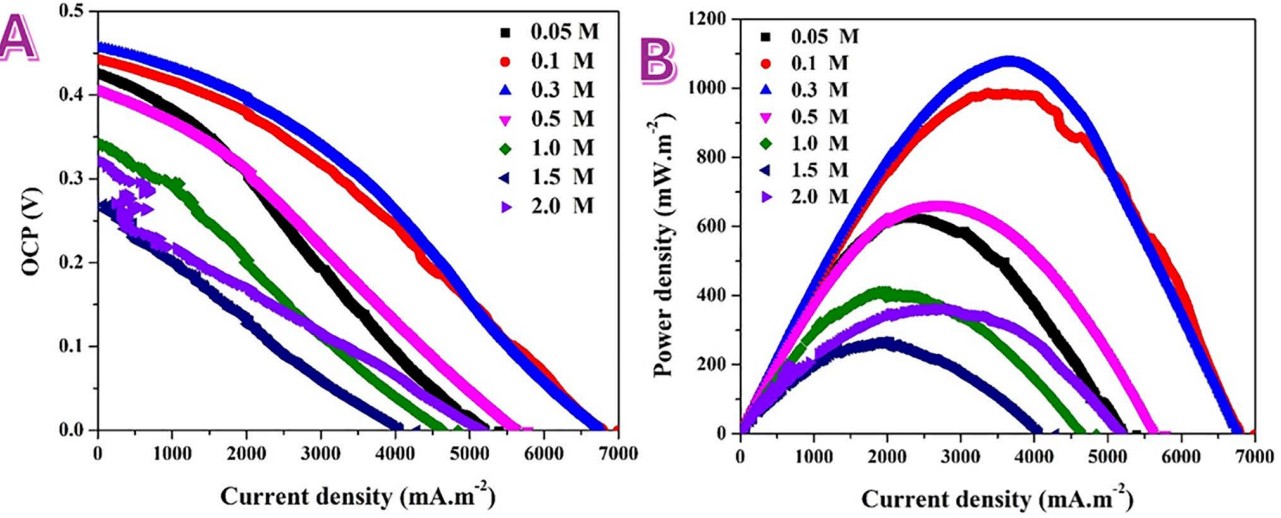

**Fig 7. Performance evaluation of membrane-less direct urea fuel cells assembled using the MAP 1 wt%-based anode at different urea concentrations.** A) Polarization curves and B) Power density curves at different urea solution concentrations for membrane-less air cathode direct urea cells assembled using MAP 1 wt%-based anode.

The power–current plots (Fig 7B) reveal a clear volcano-type dependence on urea concentration. The peak power density rises as urea concentration is increased from 0.05 to ~0.3–0.5 M, reaching ≈1.0–1.1 W.m$^{-2}$ at currents of ~3.5–4.5 A.m$^{-2}$. At still higher concentrations (≥1.0 M) the peak power drops (to a few $10^2$ mW.m$^{-2}$ at 1.5–2.0 M) because the OCP penalty and cathode losses from crossover outweigh the anode's gain in kinetic current. The position of the maximum shifts to higher current with increasing urea concentration, consistent with improved mass transport at the anode but growing mixed-potential effects at the cathode.

The respectable DUFC performance at moderate concentration is enabled by the MAP–nanofiber anode's NiOOH-active surface on a conductive carbon scaffold (XPS: $NiO_x(OH)_y$ shell, oxygenated-carbon support, phosphate anchoring), which promotes rapid OH$^-$/electron transfer and mechanical stability after the post-treatment. The decline at high concentration is a cell-level rather than catalyst-intrinsic limitation—principally crossover-induced voltage loss and ORR inhibition at the air cathode in this membrane-less geometry. Within this cell architecture and loading, ~0.3–0.5 M urea offers the best compromise between anode kinetics and cathode voltage retention.

The cell-level tests highlight how device physics can override half-cell trends. Polarization curves (Fig 8A, at 0.5 M urea solution) show that at 35 °C the membrane-less air-cathode DUFC delivers the highest open-circuit potential (≈0.45 V) and the largest limiting current to ~6.7 A.m$^{-2}$. Raising the temperature to 45 and 55 °C sharply lowers the OCP (to ≤0.15–0.20 V) and the attainable current (zero-voltage intercept drops to ~4.3 and ~3.6 A.m$^{-2}$, respectively). The accompanying power curves (Fig 8B) peak at ≈1.0 W m$^{-2}$ near 3.5–4.5 A.m$^{-2}$ at 35 °C, but fall to only ~0.15–0.20 W.m$^{-2}$ at 45–55 °C (max near 2.0–2.5 A.m$^{-2}$). Thus, increasing temperature reduces peak power by ~80–85% and shifts the optimum to lower current.

These counter-intuitive losses (given that anode kinetics usually accelerate with temperature) are characteristic of membrane-less air-cathode DUFCs and arise from cathode-dominated effects:

1. Higher temperature increases urea and NH$_3$ diffusion and volatility. Their arrival at the Pt/C air cathode drives parasitic oxidation, depressing the OCP and steepening cathodic polarization (mixed ORR/UOR).

2. Although ORR kinetics benefit from the temperature, the solubility of O$_2$ in alkaline solution decreases with temperature, pushing the cathode earlier into mass-transport limitation; the polarization curves accordingly drop more rapidly.

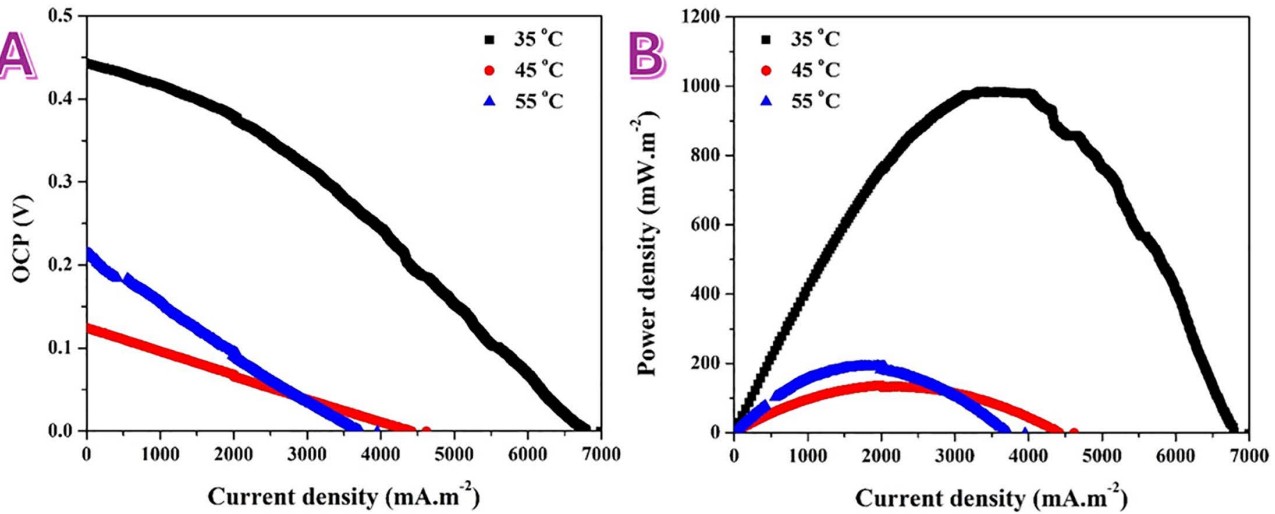

**Fig 8. Temperature-dependent performance of membrane-less direct urea fuel cells assembled using the MAP 1 wt%-based anode.** A) Polarization curves and B) Power density curves at different temperatures and 0.5 M urea solution for membrane-less air cathode direct urea cells assembled using MAP 1 wt%-based anode.

3. Heating promotes water loss from the gas-diffusion layer while simultaneously increasing vapor generation from urea oxidation; either dry-out (loss of ionic pathways) or local flooding (blocked $O_2$ access) reduces the effective ORR area.

4. Elevated $NH_3$ at higher temperature can adsorb on Pt, further lowering ORR activity.

By contrast, the anode itself remains competent at higher temperature (as confirmed in half-cell tests), so the net decline is cell-architecture-driven, not catalyst-intrinsic. Practically, these results indicate that this DUFC configuration performs best near ambient (≈35 °C). Further gains at elevated temperature would require crossover mitigation (e.g., a thin anion-exchange separator or increased cathode hydrophobicity), $NH_3$-tolerant ORR catalysts, and optimized GDL/microporous-layer design to stabilize the triple-phase boundary. The enhanced fuel cell performance is attributed to the synergistic effect of the conductive nanofiber structure and the modified surface chemistry. The fibrous network promotes efficient electron transport and mass transfer, while P and N functionalities facilitate the $Ni^{+2}/Ni^{+3}$ redox process and stabilize the NiOOH active phase, resulting in improved power output.

## 4. Conclusions

N/P-doped Ni/CNFs can be prepared by one-pot electrospinning of PVA/Ni-acetate dopes containing ammonium-phosphate additives, followed by vacuum drying (60 °C, 10 h) and carbonization. MAP and DAP can be utilized as P and N precursors, however they impart distinct morphology and surface chemistry: MAP yields uniform, well-decorated fibers with a thinner $NiO_x(OH)_y$ shell that retains a lattice-O component, on a slightly more oxygenated carbon surface; DAP drives rougher, more heavily decorated fibers comprising $Ni^0$ cores wrapped by a $Ni^{+3}$-rich, hydroxylated/phosphate-ligated shell, with less oxygenated carbon but greater surface hydroxylation (facilitating $Ni(OH)_2 \leftrightarrow NiOOH$). However, TAP is not recommended as an electrospinning additive under these conditions because it destabilizes the jet (excess solution conductivity/electrospraying), preventing continuous fiber formation. As electrocatalysts, the nanofibers show high UOR activity with composition optimization: activity and Ni-redox charge follow DAP > MAP > AP-free, with a clear volcano vs DAP loading and an optimum at 1 wt% DAP. The optimized electrode exhibits a low apparent activation energy (9.82 kJ.mol$^{-1}$), strong positive responses to urea and KOH concentrations, and 70–80% current retention over 1000 s in chronoamperometry. The nanofibers can be directly utilized as DUFC anodes; in a membrane-less air-cathode cell the MAP-based anode delivered ~1.1 W.m$^{-2}$ peak power at 35 °C (0.3–0.5 M urea), with higher-temperature losses attributed to crossover/ORR limitations rather than catalyst instability. Overall, ammonium-phosphate–assisted electrospinning provides a scalable, low-cost route to N/P-modified Ni/CNFs whose structure and surface chemistry can be tuned for efficient urea electro-oxidation and practical DUFC operation, offering clear guidance for future optimization of dopant identity/loading and cell architecture. This work demonstrates a promising strategy for the simultaneous treatment of urea-containing wastewater and generation of renewable electricity, contributing to the development of sustainable and energy-efficient environmental technologies.

## Author contributions

**Data curation:** Ahmed Saadawi.

**Formal analysis:** Nasser A. M. Barakat.

**Funding acquisition:** Gaber Edris.

**Investigation:** Ayman Yousef.

**Methodology:** Ahmed Saadawi.

**Resources:** Osama M. Erfan, Ibrahim Mustafa.

**Supervision:** Nasser A. M. Barakat.

**Writing – original draft:** Nasser A. M. Barakat.

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
