## [Decision Letter · Decision Letter 0]

11 Apr 2026

PONE-D-26-14686Interface-Engineered Nickel Oxyhydroxide on Carbon Nanofibers for Efficient Urea Oxidation and Wastewater-to-Energy ConversionPLOS One

Dear Dr. Barakat,

Thank you for submitting your manuscript to PLOS ONE. After careful consideration, we feel that it has merit but does not fully meet PLOS ONE’s publication criteria as it currently stands. Therefore, we invite you to submit a revised version of the manuscript that addresses the points raised during the review process.

We look forward to receiving your revised manuscript.

Kind regards,

Zafar Khan Ghouri

Academic Editor

PLOS One

Journal Requirements:

Additional Editor Comments:

Dear Authors,

Thank you for submitting your manuscript entitled “Interface-Engineered Nickel Oxyhydroxide on Carbon Nanofibers for Efficient Urea Oxidation and Wastewater to Energy Conversion” to PLOS One.

After careful evaluation, we have now received the reviewers’ reports. The reviewers find the topic of your work interesting and potentially suitable for publication. However, they have raised several important concerns that must be addressed before the manuscript can be considered for publication.

Based on these recommendations, the decision on your manuscript is: Minor Revision.

We invite you to revise your manuscript by carefully addressing all the reviewers’ comments and suggestions.

Your revised manuscript will be re-evaluated, and it may be sent back to the reviewers for further assessment.

We believe that addressing these concerns will significantly improve the quality and impact of your work. We look forward to receiving your revised submission.

Kind regards,

Zafar Khan Ghouri, Ph.D

Reviewer's Responses to Questions

**Comments to the Author**

1. Is the manuscript technically sound, and do the data support the conclusions?

Reviewer #1: Yes

Reviewer #2: Yes

2. Has the statistical analysis been performed appropriately and rigorously? 

Reviewer #1: Yes

Reviewer #2: Yes

3. Have the authors made all data underlying the findings in their manuscript fully available?

Reviewer #1: Yes

Reviewer #2: Yes

4. Is the manuscript presented in an intelligible fashion and written in standard English?

Reviewer #1: Yes

Reviewer #2: Yes

5. Review Comments to the Author

Reviewer #1: General comment

The manuscript presents the synthesis of ammonium-phosphate-modified Ni–carbon nanofibers and their application for urea electrooxidation and membrane-less direct urea fuel cells enabling urea remediation along with renewable power generation. The topic is relevant and the material design is interesting. However, the manuscript can be further strengthened by addressing the following points.

Specific comments

1. Clarification of the role of ammonium phosphate

The manuscript discusses the beneficial effect of MAP and DAP additives; however, the mechanistic role of phosphate and nitrogen functionalities in enhancing catalytic activity should be discussed more clearly, particularly in relation to the Ni+2/Ni+3 redox transformation and surface adsorption properties.

2. Abbreviations consistency

All abbreviations should be defined at their first appearance in both the abstract and main text, and used consistently throughout the manuscript.

3. Comparison with previous Ni-based catalysts

A brief comparison with previously reported Ni-based carbon nanofiber or Ni-based urea oxidation catalysts would help highlight the advantages of the present material in terms of activity and stability.

4. Stability discussion

The chronoamperometric stability results are promising; however, the discussion should better emphasize the structural robustness of the catalyst and relate it to the physicochemical characterization results.

5. Fuel cell discussion

The direct urea fuel cell performance is interesting, but the manuscript would benefit from a short discussion explaining how electrode structure and heteroatom doping influence the observed power density.

6. Minor language improvement

The manuscript would benefit from minor English editing to improve clarity and readability.

7. Figure clarity

Some figures could be improved by ensuring consistent font size and labeling style, particularly for electrochemical plots.

8. Conclusion section

The conclusion could be strengthened by more clearly highlighting the practical significance of converting urea-containing wastewater into electrical energy.

9. Affiliation

The affiliation of the author Ayman Youcef is missing.

Reviewer #2: The manuscript addresses an important topic in the area of low-cost electrocatalysts for alkaline fuel cell applications. The use of ammonium phosphate-modified Ni–carbon nanofibers is interesting, and the experimental work appears to be carefully carried out. The combination of structural characterization with electrochemical testing is appreciated. However, the presentation can be improved in several places, particularly in clarifying the origin of the performance differences among the studied samples and in strengthening the discussion of the electrochemical results. The manuscript may be considered for publication after the following points are addressed.

Comments

1. The Introduction gives a reasonable overview of Ni-based catalysts for urea oxidation, but the specific novelty of using ammonium phosphate as a modifier for electrospun Ni–carbon nanofibers should be emphasized more clearly. It would be useful to explain how this approach differs from other reported heteroatom-doped or phosphate-modified carbon-supported Ni systems.

2. Since both mono-ammonium phosphate (MAP) and di-ammonium phosphate (DAP) were used, the manuscript would benefit from a clearer explanation of why they lead to different catalytic behaviors. The discussion should better connect these performance differences with the characterization results, especially the surface chemical states identified by XPS.

3. The ECSA values are reported, but the manuscript should more clearly state how these values correlate with the observed oxidation currents and fuel cell performance. This would make the electrochemical interpretation more convincing.

4. The CVs recorded at different scan rates are useful, but the manuscript should comment on whether the oxidation process appears mainly diffusion-controlled, surface-controlled, or mixed-controlled based on the obtained trends.

5. The stability data are interesting, but the initial current decay observed during chronoamperometry should be briefly explained. For example, the authors may comment on whether this behavior is associated with double-layer charging, adsorption of intermediates, or surface reconstruction.

6. The direct urea fuel cell results are promising, but a short paragraph explaining how the catalyst structure and surface chemistry contribute to the obtained power output would strengthen this part of the manuscript.

7. Please ensure that all chemicals used in catalyst synthesis, electrode fabrication, and fuel cell assembly are listed clearly in the Materials section.

8. The manuscript is generally understandable, but there are a number of sentences that could be improved for grammar and clarity. In addition, the authors should check the consistency of abbreviations, symbols, and units throughout the manuscript.

6. PLOS authors have the option to publish the peer review history of their article (what does this mean?). If published, this will include your full peer review and any attached files.

Reviewer #1: No

Reviewer #2: No

---

## [Author Response · Author response to Decision Letter 1]

17 Apr 2026

Dear Prof. Zafar Ghouri

Academic Editor

Plos One

Thank you for your kind response about the manuscript [PONE-D-26-14686] titled:

“Interface-Engineered Nickel Oxyhydroxide on Carbon Nanofibers for Efficient Urea Oxidation and Wastewater-to-Energy Conversion”

The referee's comments were helpful to strength the manuscript. We would like to inform you that we have modified the manuscript according to the given comments.

To make it more easily, we have written the comments in bold phase followed by the responses in normal one. Moreover, in the revised manuscript, you can find the changes in the text in blue color.

We hope our responses cover all the comments. It will be our pleasure to respond about any more comments.

Thank you for your cooperation

Sincerely yours

Corresponding author

Nasser A. M. Barakat

Professor

Chemical Engineering Department, Minia university, Egypt

Reviewer #1:

General comment:

The manuscript presents the synthesis of ammonium-phosphate-modified Ni–carbon nanofibers and their application for urea electrooxidation and membrane-less direct urea fuel cells enabling urea remediation along with renewable power generation. The topic is relevant and the material design is interesting. However, the manuscript can be further strengthened by addressing the following points.

Response: We sincerely thank the reviewer for the positive evaluation of our work and for recognizing the relevance of the topic and the interest of the proposed material design. We appreciate the constructive suggestions provided to further improve the quality of the manuscript. All comments have been carefully considered, and the manuscript has been revised accordingly to enhance the clarity of presentation, strengthen the discussion, and better highlight the significance of the results. Detailed responses to each specific comment are provided below.

Specific comments

1. Clarification of the role of ammonium phosphate

The manuscript discusses the beneficial effect of MAP and DAP additives; however, the mechanistic role of phosphate and nitrogen functionalities in enhancing catalytic activity should be discussed more clearly, particularly in relation to the Ni+2/Ni+3 redox transformation and surface adsorption properties.

Response: We thank the reviewer for this important and insightful comment. In the revised manuscript, the discussion has been expanded to more clearly explain the mechanistic role of ammonium phosphate-derived phosphorus and nitrogen functionalities. Specifically, it is now clarified that the incorporation of phosphate species (P+5–O) and nitrogen functionalities modifies the electronic environment of the Ni active sites and facilitates the reversible Ni+2/Ni+3 redox transition, which is essential for the formation of the catalytically active NiOOH phase. In addition, the presence of phosphate groups enhances the adsorption of OH⁻ species and stabilizes surface intermediates, while nitrogen doping improves the electrical conductivity of the carbon nanofiber matrix and strengthens the interaction between Ni species and the support.

These combined effects lead to improved charge-transfer kinetics and higher electrochemical activity. The corresponding explanation has been added and supported by the XPS analysis and electrochemical results (CV redox features and activity trends) in the revised manuscript.

This is paragraph has been added to the revised manuscript

“The improved catalytic activity can be attributed to the synergistic effect of phosphorus and nitrogen functionalities introduced by ammonium phosphate. Phosphate groups facilitate OH⁻ adsorption and stabilize the NiOOH active phase, while nitrogen doping enhances electrical conductivity and strengthens metal–support interactions. These effects collectively promote the Ni²⁺/Ni³⁺ redox transformation and improve urea electrooxidation kinetics.”

2. Abbreviations consistency

All abbreviations should be defined at their first appearance in both the abstract and main text, and used consistently throughout the manuscript.

Response: We thank the reviewer for this valuable comment. The manuscript has been carefully revised to ensure that all abbreviations are defined at their first occurrence in both the abstract and the main text. In addition, a thorough check was performed to maintain consistent usage and formatting of all abbreviations throughout the manuscript.

3. Comparison with previous Ni-based catalysts.

A brief comparison with previously reported Ni-based carbon nanofiber or Ni-based urea oxidation catalysts would help highlight the advantages of the present material in terms of activity and stability.

Response: We thank the reviewer for this helpful suggestion. In the revised manuscript, a brief comparison with previously reported Ni-based carbon nanofiber and Ni-based urea oxidation catalysts has been added to better position the present material within the literature. This comparison highlights that the ammonium-phosphate-modified Ni–carbon nanofibers developed in this work combine the advantages of a conductive nanofibrous carbon network, well-dispersed Ni active sites, and phosphorus/nitrogen surface functionalities, which together contribute to improved redox reversibility, high catalytic activity, and stable electrochemical behavior.

The added discussion also clarifies that the enhanced performance of the present catalyst is associated not only with the presence of Ni active centers, but also with the synergistic role of the modified CNF support in facilitating charge transfer and stabilizing the catalytically active NiOOH phase during urea oxidation.

This is paragraph has been added to the revised manuscript

“Compared with previously reported Ni-based electrocatalysts for urea oxidation, the present ammonium-phosphate-modified Ni–carbon nanofibers offer a favorable combination of a conductive fibrous carbon framework, uniformly distributed Ni active sites, and phosphate/nitrogen surface functionalities. These features promote efficient Ni²⁺/Ni³⁺ redox transformation, enhance charge transfer, and improve catalyst stability during prolonged operation.”

4. Stability discussion

The chronoamperometric stability results are promising; however, the discussion should better emphasize the structural robustness of the catalyst and relate it to the physicochemical characterization results.

Response: We thank the reviewer for this valuable suggestion. In the revised manuscript, the discussion of the chronoamperometric stability has been expanded to better emphasize the structural robustness of the catalyst and its relation to the physicochemical characteristics. Specifically, it is now clarified that the observed stable current response is associated with the uniform dispersion of Ni nanoparticles within the partially graphitized carbon nanofiber matrix, as confirmed by TEM analysis, which provides strong physical confinement and prevents particle agglomeration.

In addition, the XPS results indicate the presence of phosphorus- and nitrogen-containing surface functionalities, which enhance metal–support interactions and contribute to stabilizing the active Ni species during repeated redox cycling. These features collectively support the durability of the catalyst under electrochemical operation. The revised text now explicitly links the stability behavior with the structural and surface properties revealed by XRD, TEM, and XPS analyses.

5. Fuel cell discussion

The direct urea fuel cell performance is interesting, but the manuscript would benefit from a short discussion explaining how electrode structure and heteroatom doping influence the observed power density.

Response: We thank the reviewer for this valuable suggestion. In the revised manuscript, a short discussion has been added to better explain the relationship between the electrode structure, heteroatom doping, and the observed fuel cell performance. It is now clarified that the interconnected carbon nanofiber network provides an efficient pathway for electron transport and facilitates mass transfer within the electrode.

In addition, the incorporation of phosphorus and nitrogen functionalities modifies the surface electronic properties of the Ni active sites, enhances the Ni+2/Ni+3 redox activity, and improves the adsorption of reactants and intermediates. These combined effects contribute to improved charge-transfer kinetics and reduced internal resistance, which are directly reflected in the enhanced power density. The corresponding explanation has been incorporated into the revised manuscript.

This paragraph has been added to the revised manuscript

“The enhanced fuel cell performance is attributed to the synergistic effect of the conductive nanofiber structure and the modified surface chemistry. The fibrous network promotes efficient electron transport and mass transfer, while P and N functionalities facilitate the Ni+2/Ni+3 redox process and stabilize the NiOOH active phase, resulting in improved power output”.

6. Minor language improvement

The manuscript would benefit from minor English editing to improve clarity and readability.

Response: We thank the reviewer for this helpful suggestion. The manuscript has been carefully revised to improve the clarity, grammar, and overall readability of the text. All sections were edited to ensure concise and clear scientific expression.

7. Figure clarity

Some figures could be improved by ensuring consistent font size and labeling style, particularly for electrochemical plots.

Response: We thank the reviewer for this helpful comment. All figures have been carefully revised to ensure consistent font size, labeling style, and overall presentation quality, particularly for the electrochemical plots. The updated figures improve clarity and readability throughout the manuscript.

8. Conclusion section

The conclusion could be strengthened by more clearly highlighting the practical significance of converting urea-containing wastewater into electrical energy.

Response: We thank the reviewer for this valuable suggestion. The Conclusion section has been revised to more clearly emphasize the practical and environmental significance of the present work. In particular, it now highlights the potential of the developed Ni–carbon nanofiber catalysts for simultaneous urea remediation and renewable electricity generation, demonstrating a sustainable approach for wastewater treatment and energy recovery.

This sentence was added to the revised manuscript

This work demonstrates a promising strategy for the simultaneous treatment of urea-containing wastewater and generation of renewable electricity, contributing to the development of sustainable and energy-efficient environmental technologies.

9. Affiliation

The affiliation of the author Ayman Youcef is missing.

Response: We thank the reviewer for pointing this out. The missing affiliation of Ayman Youcef has now been added in the revised manuscript.

Reviewer #2:

The manuscript addresses an important topic in the area of low-cost electrocatalysts for alkaline fuel cell applications. The use of ammonium phosphate-modified Ni–carbon nanofibers is interesting, and the experimental work appears to be carefully carried out. The combination of structural characterization with electrochemical testing is appreciated. However, the presentation can be improved in several places, particularly in clarifying the origin of the performance differences among the studied samples and in strengthening the discussion of the electrochemical results. The manuscript may be considered for publication after the following points are addressed.

Response: We sincerely thank the reviewer for the positive evaluation of our work and for recognizing its relevance and experimental rigor. We also appreciate the constructive suggestions aimed at improving the clarity and depth of the manuscript. In the revised version, we have carefully addressed all the points raised by the reviewer. In particular, the discussion has been expanded to better clarify the origin of the performance differences among the investigated samples, with emphasis on the role of ammonium phosphate-derived functionalities and their influence on the electrochemical behavior. Additionally, the interpretation of the electrochemical results has been strengthened to provide a clearer mechanistic understanding. Detailed responses to each specific comment are provided below.

Comments

1. The Introduction gives a reasonable overview of Ni-based catalysts for urea oxidation, but the specific novelty of using ammonium phosphate as a modifier for electrospun Ni–carbon nanofibers should be emphasized more clearly. It would be useful to explain how this approach differs from other reported heteroatom-doped or phosphate-modified carbon-supported Ni systems.

Response: We thank the reviewer for this insightful comment. In the revised manuscript, the Introduction has been expanded to more clearly highlight the novelty of using ammonium phosphate as a multifunctional additive in the electrospinning precursor. Unlike conventional approaches that rely on post-synthesis doping or the introduction of separate heteroatom sources, the present method enables the simultaneous incorporation of phosphorus and nitrogen functionalities during nanofiber formation and carbonization, leading to a more homogeneous distribution of dopants and strong interaction with Ni species.

It is now clarified that this approach differs from previously reported systems by combining (i) in-situ P,N co-doping, (ii) controlled dispersion of Ni nanoparticles within a conductive nanofibrous carbon matrix, and (iii) direct tuning of the Ni²⁺/Ni³⁺ redox behavior through surface chemistry modification. These features collectively contribute to improved catalytic activity and stability. The corresponding explanation has been incorporated into the Introduction to better distinguish the present work from existing Ni-based and heteroatom-doped catalyst systems.

This sentence has been added to the revised manuscript

In contrast to conventional heteroatom-doping strategies, the use of ammonium phosphate in the electrospinning precursor enables in-situ incorporation of both phosphorus and nitrogen functionalities, leading to a homogeneous distribution of dopants and enhanced interaction with Ni active sites

2. Since both mono-ammonium phosphate (MAP) and di-ammonium phosphate (DAP) were used, the manuscript would benefit from a clearer explanation of why they lead to different catalytic behaviors. The discussion should better connect these performance differences with the characterization results, especially the surface chemical states identified by XPS.

Response: We thank the reviewer for this important comment. In the revised manuscript, the discussion has been expanded to provide a clearer explanation of the different catalytic behaviors observed for MAP- and DAP-modified samples. It is now clarified that the difference arises from the distinct phosphorus and nitrogen environments introduced by MAP and DAP, which affect the surface chemistry and electronic structure of the catalyst.

In particular, the XPS analysis shows variations in the relative contributions of phosphate (P⁵⁺–O) species and nitrogen functionalities, which influence the interaction between Ni active sites and the carbon matrix. These differences impact the Ni+2/Ni+3 redox transition, the formation and stabilization of the NiOOH active phase, and the adsorption of reaction intermediates during urea oxidation.

The revised discussion now explicitly correlates the observed electrochemical performance trends with the XPS results, providing a more consistent structure–property relationship between dopant chemistry and catalytic activity.

This sentence has been added to the revised manuscript

“The difference in catalytic performance between MAP- and DAP-modified samples can be attributed to their distinct surface chemical environments, as revealed by XPS. Variations in phosphate and nitrogen functionalities influence the interaction with Ni active sites, thereby affecting the Ni²⁺/Ni³⁺ redox behavior and the stability of the NiOOH phase during electrooxidation.”

3. The ECSA values are reported, but the

---

## [Editor Report · Decision Letter 1]

21 Apr 2026

Interface-Engineered Nickel Oxyhydroxide on Carbon Nanofibers for Efficient Urea Oxidation and Wastewater-to-Energy Conversion

PONE-D-26-14686R1

Dear Dr. Barakat,

We’re pleased to inform you that your manuscript has been judged scientifically suitable for publication and will be formally accepted for publication once it meets all outstanding technical requirements.

Kind regards,

Zafar Khan Ghouri

Academic Editor

PLOS One

Additional Editor Comments (optional):

Dear Prof. Barakat,

Thank you for submitting the revised version of your manuscript entitled “Interface-Engineered Nickel Oxyhydroxide on Carbon Nanofibers for Efficient Urea Oxidation and Wastewater-to-Energy Conversion” (Manuscript ID: PONE-D-26-14686R1).

We are pleased to inform you that, following the first round of peer review and assessment of your revisions, your manuscript has been accepted for publication. The reviewers’ comments have been adequately addressed, and the manuscript meets the journal’s standards for publication.

Yours sincerely,

Zafar Khan Ghouri, Ph.D.
---

## [Editor Report · Acceptance letter]

PONE-D-26-14686R1

PLOS One

Dear Dr. Barakat,

I'm pleased to inform you that your manuscript has been deemed suitable for publication in PLOS One. Congratulations! Your manuscript is now being handed over to our production team.

Kind regards,

on behalf of

Dr. Zafar Ghouri

Academic Editor

PLOS One